# CDK-regulated dimerization of M18BP1 on a Mis18 hexamer is necessary for CENP-A loading

Dongqing Pan[1]*, Kerstin Klare[1†], Arsen Petrovic[1], Annika Take[1], Kai Walstein[1], Priyanka Singh[1], Arnaud Rondelet[1], Alexander W Bird[1], Andrea Musacchio[1,2]*

[1]Department of Mechanistic Cell Biology, Max-Planck Institute of Molecular Physiology, Dortmund, Germany; [2]Centre for Medical Biotechnology, Faculty of Biology, University Duisburg-Essen, Essen, Germany

**Abstract** Centromeres are unique chromosomal loci that promote the assembly of kinetochores, macromolecular complexes that bind spindle microtubules during mitosis. In most organisms, centromeres lack defined genetic features. Rather, they are specified epigenetically by a centromere-specific histone H3 variant, CENP-A. The Mis18 complex, comprising the Mis18$\alpha$:Mis18$\beta$ subcomplex and M18BP1, is crucial for CENP-A homeostasis. It recruits the CENP-A-specific chaperone HJURP to centromeres and primes it for CENP-A loading. We report here that a specific arrangement of Yippee domains in a human Mis18$\alpha$:Mis18$\beta$ 4:2 hexamer binds two copies of M18BP1 through M18BP1's 140 N-terminal residues. Phosphorylation by Cyclin-dependent kinase 1 (CDK1) at two conserved sites in this region destabilizes binding to Mis18$\alpha$:Mis18$\beta$, limiting complex formation to the G1 phase of the cell cycle. Using an improved viral 2A peptide co-expression strategy, we demonstrate that CDK1 controls Mis18 complex recruitment to centromeres by regulating oligomerization of M18BP1 through the Mis18$\alpha$:Mis18$\beta$ scaffold.

*For correspondence: dongqing. pan@mpi-dortmund.mpg.de (DP); andrea.musacchio@mpi-dortmund.mpg.de (AM)

Present address: [†]Department of Molecular Genetics, Faculty of Biology, Centre for Medical Biotechnology, University of Duisburg-Essen, Essen, Germany

## Introduction

In all eukaryotes, faithful chromosome duplication and segregation to the daughter cells is essential for cell viability and development. During mitosis, chromosomes bi-orient on the mitotic spindle, a structure made of microtubules, microtubule motors, and microtubule-binding proteins (*Heald and Khodjakov, 2015*). Chromosomes attach to spindle microtubules through large multisubunit assemblies known as kinetochores (*Pesenti et al., 2016*). Besides microtubule binding, kinetochores harness a poorly understood molecular mechanism of 'error correction' that leads to selective stabilization of bi-oriented attachments (*Foley and Kapoor, 2013*). Upon completion of bi-orientation, removal of sister chromatid cohesion allows the separation of the sister chromatids to the two newly forming daughter cells.

How kinetochores assemble, and how their position on chromosomes is maintained generation after generation, is an important and partly unresolved challenge. Kinetochores assemble on centromeres, unique chromosomal loci characterized by a strong enrichment of the histone H3 variant centromere protein A (CENP-A) (*Earnshaw and Rothfield, 1985*). Centromeres in different organisms often consist of long arrays of repetitive sequences, such as the human 171-basepair $\alpha$-satellite repeats (*Fukagawa and Earnshaw, 2014*; *McKinley and Cheeseman, 2016*). However, centromeres are often found associated with non-repetitive and non-evolutionarily conserved sequences in different organisms. Furthermore, there are instances in humans and other species of neo-centromeres that become established on non-repetitive sequences (*Fukagawa and Earnshaw, 2014*;

*McKinley and Cheeseman, 2016*). These examples indicate that the sequence of centromeric DNA contributes only marginally to centromere identity.

There is now general agreement that CENP-A itself is an epigenetic factor for specification of centromeres (*Cleveland et al., 2003*; *De Rop et al., 2012*; *Stellfox et al., 2013*). CENP-A is required for the recruitment of several inner (centromere-proximal) kinetochore proteins, now generally referred to as constitutive centromere associated network (CCAN) (*Foltz et al., 2006*; *Hori et al., 2008*; *Izuta et al., 2006*; *Obuse et al., 2004*; *Okada et al., 2006*), and thus acts as the kinetochore's foundation. CENP-A, together with other CCAN proteins, also promotes the recruitment of specialized machinery devoted to its own incorporation in chromatin (*Fukagawa and Earnshaw, 2014*; *McKinley and Cheeseman, 2016*). This limits incorporation of new CENP-A to the position of the existing CENP-A domain, thus preserving centromere identity. Importantly, the amount of CENP-A at centromeres is halved during DNA replication, and the ensuing CENP-A 'vacancy' may be filled with histone H3.3 (*Dunleavy et al., 2011*). The reduction in CENP-A is compensated with new CENP-A deposition after cell division, in the G1 phase of the cell cycle (*Bodor et al., 2014*; *Jansen et al., 2007*; *Shelby et al., 2000*), likely in a reaction that replaces H3.3 with CENP-A.

The Mis18 complex is an evolutionarily conserved functional unit of the CENP-A loading machinery (*Fujita et al., 2007*; *Hayashi et al., 2014*, *2004*). Vertebrate Mis18 complex consists of Mis18α and Mis18β, which are non-redundant and physically interacting paralogs, and M18BP1 (also known as KNL2) (*Fujita et al., 2007*). The Mis18 complex interacts with the CENP-A-specific chaperone HJURP (*Dunleavy et al., 2009*; *Foltz et al., 2009*; *Nardi et al., 2016*; *Wang et al., 2014*) and with CCAN subunits (*Dambacher et al., 2012*; *Hori et al., 2013*; *Moree et al., 2011*; *Shono et al., 2015*; *Stellfox et al., 2016*) (*Figure 1A*). Interactions of this machinery enable the recruitment of new CENP-A to already existing centromeres.

M18BP1 is an 1132-residue protein whose sequence is predicted to be largely unstructured. It contains a conserved 50-residue SANT (Swi3, Ada2, N-CoR, and TFIIIB) domain and a 100-residue SANT-associated (SANTA) domain (*Boyer et al., 2004*; *Fujita et al., 2007*; *Maddox et al., 2007*; *Zhang et al., 2006*) (*Figure 1B*). The high content of unstructured regions suggests that M18BP1 functions as a hub for multiple protein-protein interactions. Indeed, CENP-C, CENP-I, Polo-like kinase 1 (Plk1), MgcRacGAP, and the histone acetyl transferase (HAT) KAT7 are suggested to interact with M18BP1 (*Dambacher et al., 2012*; *Lagana et al., 2010*; *McKinley and Cheeseman, 2014*; *Moree et al., 2011*; *Ohzeki et al., 2016*; *Shono et al., 2015*), but the molecular details of these interactions remain poorly defined.

On the other hand, significant progress has been made in the characterization of the interaction of M18BP1 with Mis18α and Mis18β. Mis18α and Mis18β have similar domain structures, with an N-terminal unstructured region, a Yippee domain in the middle region, and a C-terminal coiled-coil. Recent studies of the single *Schizosaccharomyces pombe* Mis18 ortholog and of the human Mis18α: Mis18β complex suggested that Mis18 proteins form tetramers (*Nardi et al., 2016*; *Subramanian et al., 2016*). A segment of M18BP1 comprising ~380 N-terminal residues was shown to be responsible for a physical interaction with the Mis18α:Mis18β complex (*Ohzeki et al., 2016*; *Stellfox et al., 2016*).

The assembly of the CENP-A deposition machinery in the G1 phase is regulated by inhibitory CDK phosphorylation of CENP-A (*Yu et al., 2015*), HJURP (*Müller et al., 2014*; *Wang et al., 2014*), and M18BP1 (*McKinley and Cheeseman, 2014*; *Silva et al., 2012*). Moderate CDK activity through the S and G2 phases and high CDK activity in M phase of the cell cycle keep the CENP-A loading machinery disassembled (*Silva et al., 2012*). Degradation of Cyclin B at anaphase and the ensuing decline in CDK activity reverts this condition, allowing physical interactions of the CENP-A loading machinery. The Mis18 complex starts localizing to the CENP-A domain from anaphase and recruits HJURP and new CENP-A in early G1 phase (*Dunleavy et al., 2009*; *McKinley and Cheeseman, 2014*; *Nardi et al., 2016*). Functionally relevant CDK phosphorylation sites in CENP-A and HJURP were identified (*Müller et al., 2014*; *Yu et al., 2015*), but those in M18BP1 (show in *Figure 1—figure supplement 1*) remain functionally uncharacterized.

In this study, we report that two copies of M18BP1 bind an hexameric Mis18α:Mis18β complex, and that dimerization of M18BP1 is important for its recruitment to centromeres. We show that M18BP1 binds Mis18α:Mis18β through two sub-regions in its N-terminal 140 amino acids. Single CDK phosphorylation sites in each sub-region, Thr40 and Ser110, regulate the interaction, with

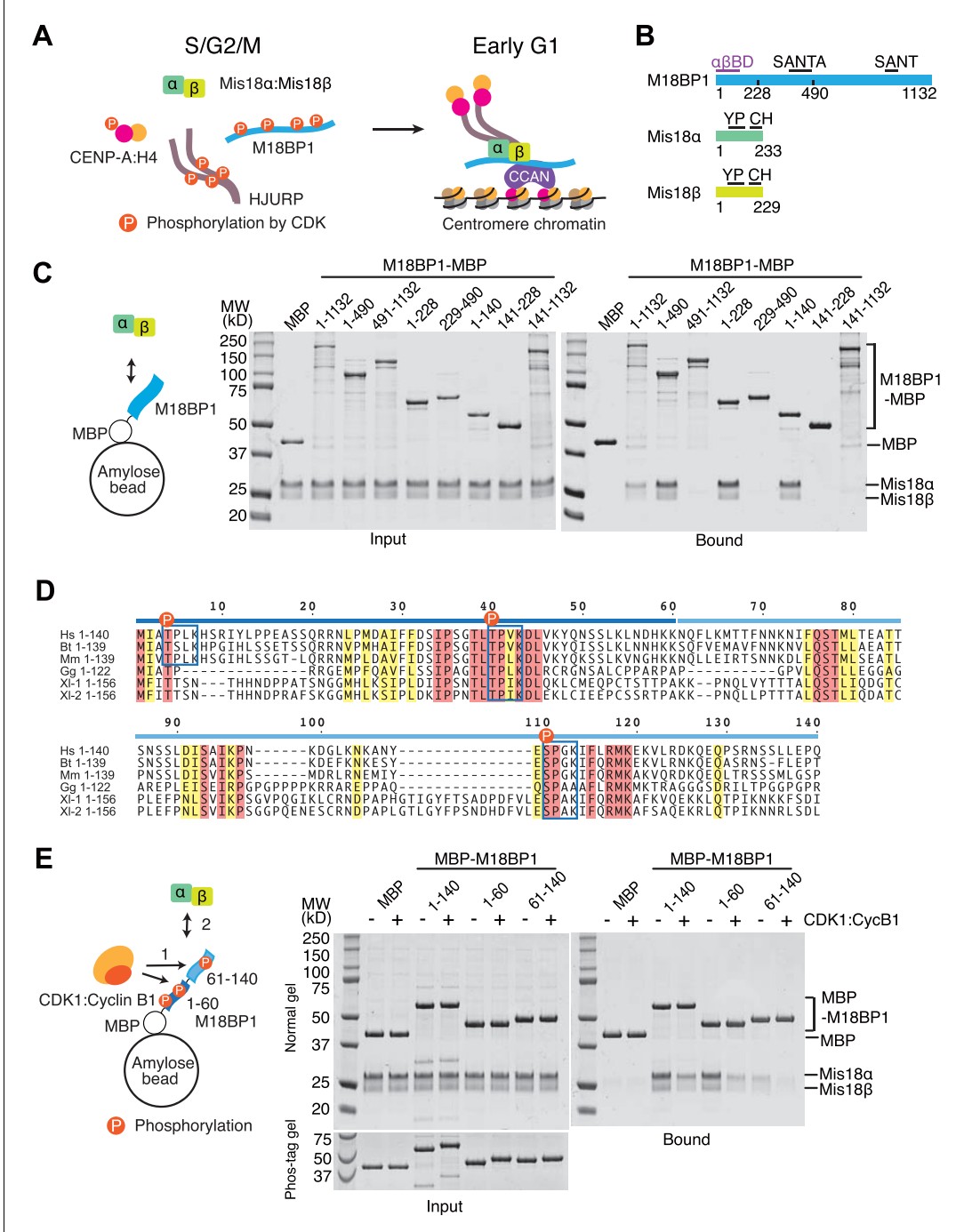

**Figure 1.** CDK1 regulates binding of M18BP1$^{1-140}$ and Mis18α:Mis18β. (**A**) Simplified current model of the cell-cycle-dependent assembly of the CENP-A deposition machinery. CDKs phosphorylate CENP-A, HJURP, and M18BP1 to prevent the assembly of the protein complex during the S, G2, and M phases of the cell cycle. Reduced CDK activity in early G1 phase allows the CENP-A deposition machinery to interact and localize to centromeres through the interaction with subunits of CCAN. (**B**) Schematic diagrams of domain structures of M18BP1, Mis18α, and Mis18β. YP, YP domain; CH, C-terminal helix; αβBD, Mis18α:Mis18β binding domain. (**C**) Amylose-resin pull-down assays to identify the Mis18α:Mis18β-binding domain of M18BP1. M18BP1-MBP variants and MBP were produced in bacteria and Mis18α:Mis18β was produced using baculovirus co-expression system. Shown is a representative gel of pull-down assays that were repeated at least three times. The same applies to all other pull-down assays in this paper. (**D**) Sequence alignment of M18BP1$^{1-140}$. Hs, *Homo sapiens*; Mm, *Mus musculus*; Bt, *Bos taurus*; Gg, *Gallus gallus*; Xl-1, *Xenopus laevis* isoform 1; Xl-2, *Xenopus laevis* isoform 2. Blue-boxed regions indicate putative CDK phosphorylation motifs. Residues that are identical in all sequences are shaded red, and residues that only have conserved substitutions are shaded yellow. (**E**) Amylose-resin pull-down assays examining how phosphorylation affects complex formation. MBP-M18BP1 variants were incubated with or without CDK1:Cyclin B1 at 30°C for 2 hr before mixing with Mis18α:Mis18β.
*Figure 1 continued on next page*

*Figure 1 continued*

The Phos-tag containing acrylamide gel was used to detect the mobility shift caused by phosphorylation. Gels in panels C and E were stained with CBB.

The following figure supplement is available for figure 1:

**Figure supplement 1.** Previously identified phosphorylation sites on M18BP1.

phosphorylation largely reducing the affinity of M18BP1 for Mis18α:Mis18β, and phosphomimetic mutations abolishing the CENP-A loading activity of M18BP1. Thus, our results identify a mechanism to limit CENP-A loading to the G1 phase of the cell cycle.

## Results

### M18BP1$^{1–140}$ contains two sequential binding regions for Mis18α: Mis18β

We performed amylose-resin pull-down assays with purified M18BP1-MBP (maltose binding protein) fusion variants and Mis18α:Mis18β to identify the M18BP1 sequence responsible for this interaction. The N-terminal 140 residues of M18BP1 were sufficient for Mis18α:Mis18β binding (*Figure 1C*. See *Supplementary file 1* for a list of constructs used in this study), thus narrowing down the binding site for the Mis18α:Mis18β complex within the N-terminal region of M18BP1 (*Ohzeki et al., 2016*; *Stellfox et al., 2016*). A sequence alignment of vertebrate M18BP1 (*Figure 1D*) revealed that the N-terminal 140 residues of M18BP1 contain two highly conserved CDK phosphorylation motifs at positions Thr40 and Ser110 that are surrounded by other conserved residues, and a less conserved CDK motif at position Thr4. Divergent sequences around residue 50–70 of M18BP1 create a gap separating the two conserved regions. To identify the region responsible for M18BP1 binding, we split M18BP1$^{1–140}$ into two fragments (1–60 and 61–140). Both regions retained the ability to bind Mis18α:Mis18β, indicating that each region retains measurable binding affinity for Mis18α:Mis18β, with M18BP1$^{1–60}$ binding apparently more strongly than M18BP1$^{61–140}$ (*Figure 1E*). We performed pull-down assays with M18BP1 fragments that had been previously phosphorylated with recombinant human CDK1:Cyclin B1. Complete phosphorylation of the fragments was confirmed by mobility shift of the bands on Phos-tag gels. All phosphorylated M18BP1 fragments showed reduced binding affinity for Mis18α:Mis18β compared to the non-phosphorylated fragments (*Figure 1E*).

### CENP-A loading assay in HeLa cells

We used the CRISPR/Cas9 system to create an in-frame 3' fusion of the SNAP-tag coding sequence with the endogenous CENP-A coding sequence in a HeLa Flp-In-T-REx cell line (*Tighe et al., 2008*). The resulting HeLa CENP-A-SNAP cell line allowed us to perform SNAP-pulse-labeling experiments to examine new CENP-A loading at centromeres (*Jansen et al., 2007*). HeLa CENP-A-SNAP cells were treated with M18BP1 siRNA (*Fujita et al., 2007*) for 48 hr and synchronized at different phases of the cell cycle using thymidine (G1/S transition), S-trityl-L-cysteine (STLC, mitosis), or mitotic-shake-off. Existing CENP-A-SNAP at the G1/S transition phase after thymidine treatment was blocked using SNAP-Cell Block reagent. After release in the cell cycle, newly loaded CENP-A-SNAP was eventually labeled after cells had transited through mitosis using SNAP-Cell 647-SiR in early G1 phase (*Figure 2A*). Cells treated with M18BP1 siRNA showed clear CENP-A loading defects, whereas control cells showed effective CENP-A loading (*Figure 2E*, *Figure 2—figure supplement 1A*). Depletion of M18BP1 siRNA was quantitated using immunofluorescence from endogenous M18BP1 and Western blotting (*Figure 2—figure supplement 1B–C*).

### Efficient co-expression with a modified 2A-peptide strategy

Next, we wished to build an approach for testing different segments of M18BP1 for their ability to rescue kinetochore localization of Mis18α:Mis18β and CENP-A-SNAP incorporation in cells depleted of endogenous M18BP1. For this, we modified the 2A-peptide co-expression system to allow co-expression of EGFP-M18BP1 segments and of mCherry-Mis18α. 2A-peptides are 20-amino-acid viral

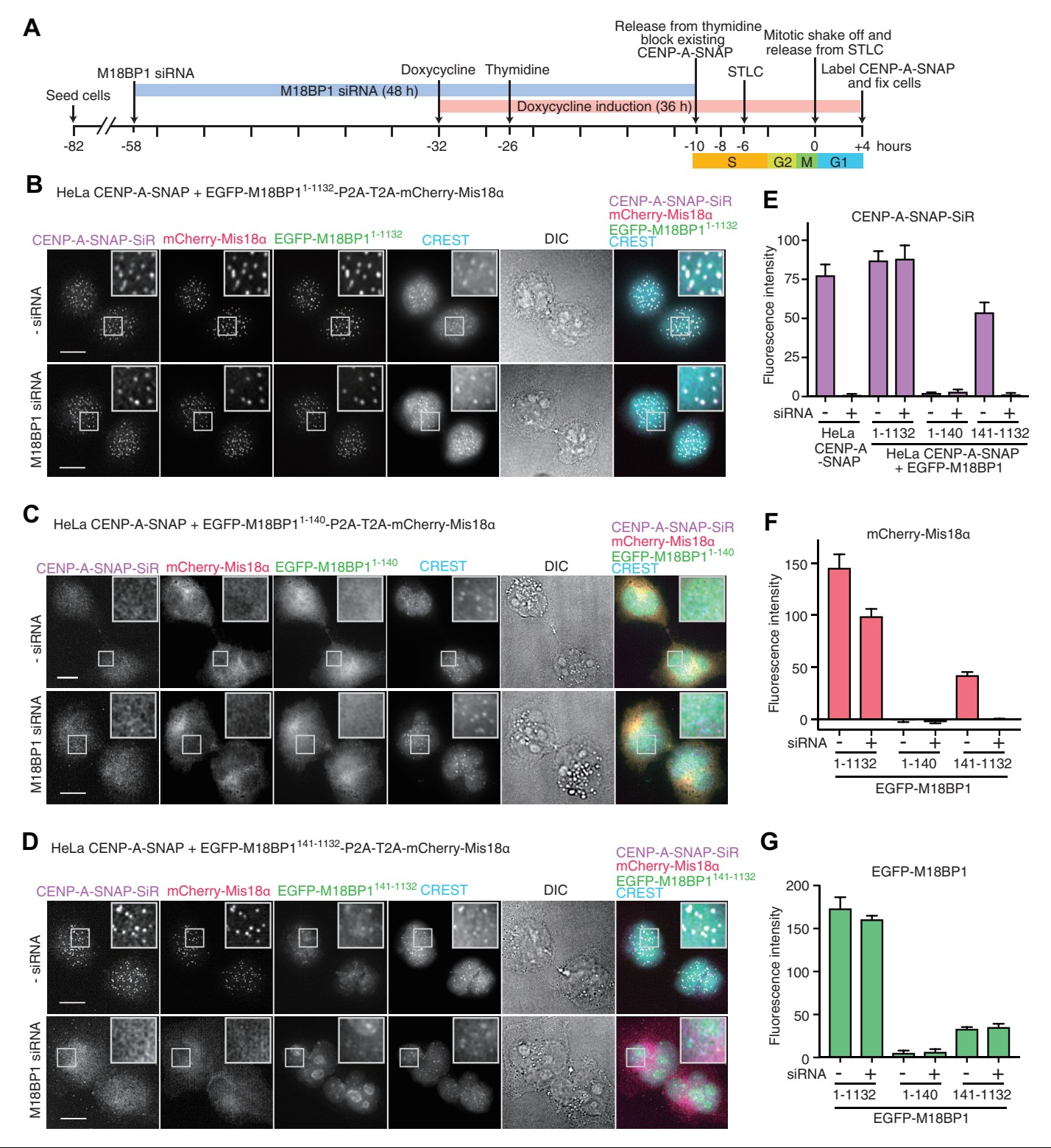

**Figure 2.** Functional analysis of M18BP1$^{1-140}$ and M18BP1$^{141-1132}$ in CENP-A recruitment. (**A**) Schematic description of the experimental procedure for testing the functionality of M18BP1 variants for new CENP-A deposition and the recruitment of Mis18α to centromeres. SNAP-Cell Block was used to block existing CENP-A-SNAP proteins at the time point −10 hr, and SNAP-Cell 647-SiR was used to label newly produced CENP-A-SNAP at the time point +4 hr. (**B–D**) Representative images showing the fluorescence of CENP-A-SNAP labeled with SNAP-Cell 647-SiR (CENP-A-SNAP-SiR), mCherry-Mis18α and EGFP-M18BP1 variants in fixed HeLa cells treated as described in panel **A**. Centromeres were visualized with CREST sera. Control cells

*Figure 2 continued on next page*

*Figure 2 continued*

were treated with transfection reagent (Lipofectamine RNAiMAX) in the absence of M18BP1 siRNA. Scale bars represent 10 μm. DIC, differential interference contrast. All cell biological experiments in this paper were repeated at least three times. (E) Quantification of the fluorescence intensity of CENP-A-SNAP-SiR on each centromere. Centromere spots were detected using images of CREST channel with the software Fiji (*Schindelin et al., 2012*). Mean intensity of CENP-A-SNAP-SiR fluorescence on centromeres was obtained from every experiment (>100 centromere spots from 10–12 early G1 cells). The bar graph presents mean values from three independent experiments. Error bars indicate SEM. (F, G) Quantification of the fluorescence intensity of mCherry-Mis18α and EGFP-M18BP1 variants on each centromere.

The following figure supplements are available for figure 2:

**Figure supplement 1.** HeLa CENP-A-SNAP cell lines.

**Figure supplement 2.** Tandem 2A-peptide system enables efficient co-expression of multiple proteins in HeLa cells.

peptides possessing 'ribosome skipping' activity and are widely used to co-express proteins in eukaryotic cells when introduced between coding sequences (*de Felipe et al., 2006*; *Kim et al., 2011*). P2A (Porcine Teschovirus-1 2A) and T2A (Thosea asigna virus 2A) are the most efficient of the known 2A-peptide sequences. However, both peptides used individually still result in incomplete 'ribosome skipping' (*Kim et al., 2011*) (*Figure 2—figure supplement 2A*), and even 5–10% unseparated fusion proteins could mislead the interpretation of co-localization experiments.

To overcome this intrinsic limitation of 2A-peptides, we connected P2A and T2A in tandem (*Figure 2—figure supplement 2A*). In test experiments, co-expression of EGFP-NLS and mCherry-PTS1 (directed to the nucleus and peroxisomes, respectively) with single P2A or T2A sequence separators resulted in a significant fraction of unseparated proteins and co-localization (*Figure 2—figure supplement 2B–D*). Co-expression with a tandem P2A-T2A construct, however, led to undetectable unseparated protein, and almost complete separation of the EGFP and mCherry signals in fluorescence microscopy experiments (*Figure 2—figure supplement 2B–D*). The same encouraging results were observed in a ternary expression that included an additional mitochondrial marker (*Figure 2—figure supplement 2E*).

We therefore fused a P2A-T2A tandem sequence in frame between full length EGFP-M18BP1 (EGFP-M18BP1$^{1-1132}$) and mCherry-Mis18α. This rescue construct was stably integrated by Flp-In recombination into the CENP-A-SNAP HeLa Flp-In-T-REx cells. Doxycycline-induced expression of EGFP-M18BP1$^{1-1132}$ rescued CENP-A-SNAP loading in cells depleted of endogenous M18BP1 (*Figure 2B,E*). In the same cells, mCherry-Mis18α showed centromere localization in G1 phase, indicating that EGFP-M18BP1$^{1-1132}$ was also able to rescue Mis18α localization (*Figure 2B and F*).

## Dominant-negative effect of M18BP1$^{1-140}$ overexpression

We examined the effects of expressing the Mis18α:Mis18β-binding region of M18BP1 in the CENP-A-SNAP loading assay. HeLa cells expressing EGFP-M18BP1$^{1-140}$ showed normal morphology but were unable to load new CENP-A-SNAP onto centromeres, regardless of whether endogenous M18BP1 had been depleted or not (*Figure 2C and E*). Both EGFP-M18BP1$^{1-140}$ and mCherry-Mis18α failed to localize to centromeres, suggesting that EGFP-M18BP1$^{1-140}$ has a strong dominant-negative effect on the function of endogenous M18BP1 (*Figure 2C, F and G*).

## M18BP1$^{141-1132}$ localizes to centromeres but is not sufficient for new CENP-A incorporation

We also examined the functionality of the M18BP1$^{141-1132}$ construct, which lacks the Mis18α:Mis18β-binding region. HeLa cells expressing EGFP-M18BP1$^{141-1132}$ showed a similar level of CENP-A-SNAP loading activity to the HeLa cells expressing EGFP-M18BP1$^{1-1132}$ in presence of endogenous M18BP1, indicating that EGFP-M18BP1$^{141-1132}$ does not exercise strong dominant-negative effects on endogenous M18BP1. However, no CENP-A-SNAP loading was observed when M18BP1 had been depleted, indicating that EGFP-M18BP1$^{141-1132}$ cannot replace endogenous M18BP1 (*Figure 2D and E*). EGFP-M18BP1$^{141-1132}$ weakly localized to centromeres, regardless of the depletion of endogenous M18BP1. However, no mCherry-Mis18α recruitment to the centromere was

observed in presence of EGFP-M18BP1$^{141–1132}$ (upon depletion of endogenous M18BP1), indicating that the interaction with M18BP1 is required for recruitment of Mis18α:Mis18β (*Figure 2D and G*).

## Phosphorylation on M18BP1 Thr40 and Ser110 prevents Mis18α:Mis18β binding

To confirm that Thr4, Thr40, and Ser110 are the major CDK-phosphorylation sites, we generated a series of non-phosphorylatable M18BP1 mutants by substituting Thr4, Thr40, and Ser110 with either Val or Ala and performed pull-down assays with Mis18α:Mis18β (*Figure 3A*). CDK1-treated T4V and T40V mutants of M18BP1$^{1-60}$ showed reduced mobility shift, and a T4V/T40V double mutant of M18BP1$^{1–60}$ showed no shift on Phos-tag gels, suggesting Thr4 and Thr40 are the major phosphorylation sites of M18BP1$^{1–60}$. The interaction between M18BP1$^{1–60/T4V}$ and Mis18α:Mis18β was sensitive to CDK-phosphorylation, while the interaction between M18BP1$^{1–60/T40V}$ and Mis18α:Mis18β was not affected by CDK activity, indicating that phosphorylation of Thr40 is a determinant of the interaction. Applying the same strategy to the third site, Ser110, we found it to be the only major phosphorylation site within M18BP1$^{61–140}$. We observed reduced affinity of M18BP1$^{61–140/S110A}$ for Mis18α:Mis18β, but the interaction between M18BP1$^{61–140/S110A}$ and Mis18α:Mis18β was clearly observed in the buffer containing 100 mM NaCl and was not affected by CDK activity. Combining T40V and S110A mutations, we were able to generate a mutant, M18BP1$^{1–140/T40V/S110A}$, which was resistant to CDK activity (*Figure 3A*).

We then generated a series of phosphomimetic-mutant M18BP1 fragments by substituting Thr4, Thr40, and Ser110 with either Asp or Glu. T4D and T4E mutants of M18BP1$^{1–60}$ bound Mis18α:Mis18β as efficiently as the wild-type construct. T40E mutant of M18BP1$^{1–60}$ showed reduced affinity for Mis18α:Mis18β, while the T40D mutant was unable to bind Mis18α:Mis18β. An S110E mutant of M18BP1$^{61–140}$ showed stronger reduction of binding affinity for Mis18α:Mis18β than an S110D mutant. Combining the T40D and S110E mutations, we observed a reduction of the affinity of M18BP1$^{1–140/T40D/S110E}$ for Mis18α:Mis18β comparable to that observed in presence of CDK1:Cyclin B1 (*Figures 3B* and *1E*).

Using the HeLa CENP-A-SNAP cell line, we determined that the effects of the M18BP1$^{1–1132/T40D/S110E}$ on localization and CENP-A incorporation were similar to those observed in presence of EGFP-M18BP1$^{141–1132}$. In both lines, we observed normal CENP-A-SNAP loading and localization of mCherry-Mis18α to centromeres in presence of endogenous M18BP1, but no CENP-A-SNAP loading nor localization of mCherry-Mis18α upon depletion of endogenous M18BP1 (*Figures 2D–G*, *4A and C–E*). When phosphomimetic mutations were introduced in the Mis18α:Mis18β-binding region (M18BP1$^{1–140/T40D/S110E}$), the strong dominant-negative effects of M18BP1$^{1–140}$ on CENP-A loading and mCherry-Mis18α localization were bypassed, suggesting that M18BP1$^{1–140/T40D/S110E}$ cannot interact with Mis18α:Mis18β (*Figures 2C, E–G*, *4B and C–E*). In agreement with these findings and with our interpretation that M18BP1$^{1–140}$ exercises a dominant-negative effect on CENP-A loading by binding to the the Mis18α:Mis18β complex complex, co-immunoprecipitation (co-IP) experiments with anti-GFP antibody coupled beads demonstrated that EGFP-M18BP1$^{1–140}$, but not EGFP-M18BP1$^{1–140/T40D/S110E}$, interacts with the Mis18α:Mis18β complex complex in HeLa cells (*Figure 4F*).

## The Mis18α:Mis18β complex is a hexamer

We established a baculovirus system for co-expression of the Mis18α and Mis18β subunits and purified the complex to homogeneity. The Mis18α:Mis18β complex appeared monodisperse in size-exclusion chromatography (SEC) profiles (*Figure 5A*). We analyzed its stoichiometry with both sedimentation velocity and sedimentation equilibrium analytical ultracentrifugation (AUC) (*Figure 5B–D*). We obtained molecular weight (MW) estimates of about 150 kD for the Mis18α:Mis18β complex from both AUC methods (*Figure 5B–D*). This corresponds to the predicted MW of a hexameric complex of Mis18α and Mis18β, but the stoichiometry remains unclear due to the similar size of the Mis18α and Mis18β subunits. To determine the stoichiometry of Mis18α and Mis18β within the complex, we tagged either Mis18α or Mis18β with MBP and purified two different complexes, 6His-Mis18α:MBP-Mis18β or MBP-Mis18α:6His-Mis18β (*Figure 5C*). The presence of each MBP in the complex increases the molecular mass of the complex by 42 kDa. The AUC results indicated unequivocally that 6His-Mis18α:MBP-Mis18β complex contains two MBPs and that MBP-

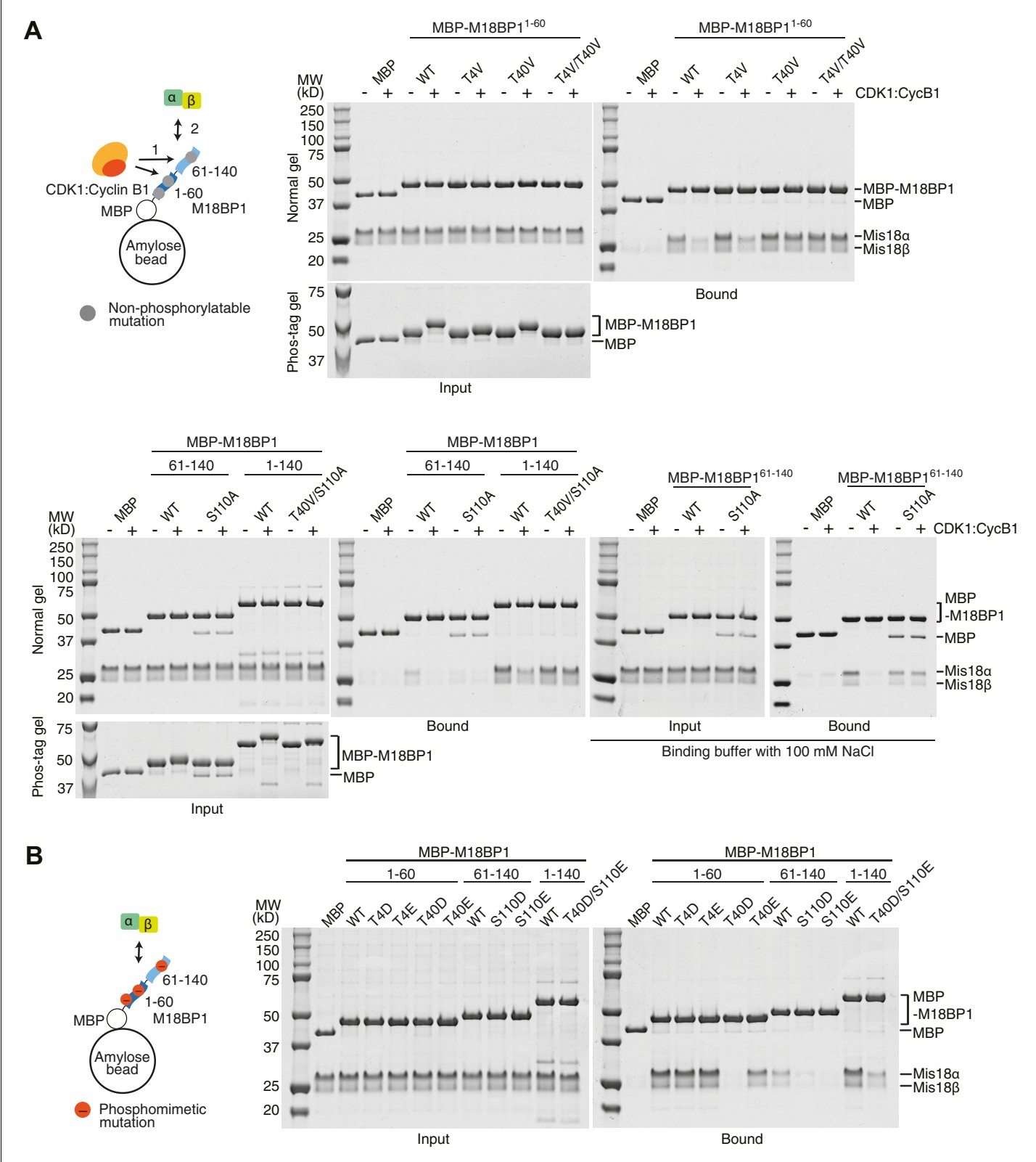

**Figure 3.** Phosphorylation on M18BP1 Thr40 and Ser110 reduces affinity for Mis18α:Mis18β. (**A, B**) Amylose-resin pull-down assays examining complex formation of Mis18α:Mis18β with non-phosphorylatable (panel **A**) or phosphomimetic (panel **B**) M18BP1 mutants. The experiments were performed as in *Figure 1C,E*. Binding buffer with 300 mM NaCl was used, unless indicated. WT, wild type.

Biochemistry | Cell Biology

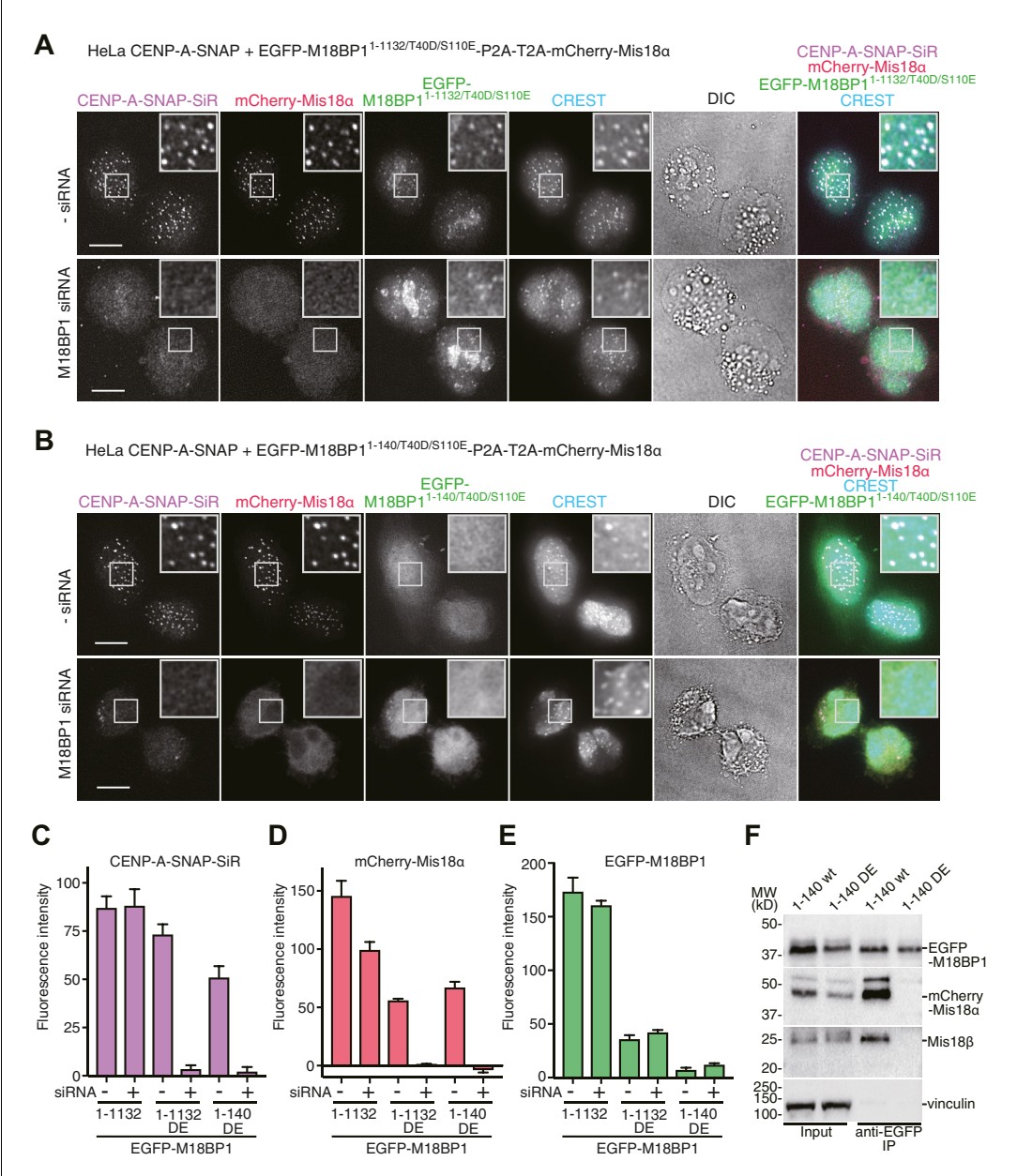

**Figure 4.** Functional analyses of M18BP1 T40D/S110E mutants in CENP-A recruitment. (**A**, **B**) Representative images showing the fluorescence of CENP-A-SNAP-SiR, mCherry-Mis18α, and EGFP-M18BP1 variants in fixed HeLa cells treated as described in *Figure 2A*. Centromeres were visualized with CREST sera. Scale bars represent 10 μm. (**C–E**) Quantification of the fluorescence intensity of CENP-A-SNAP-SiR (panel **C**), mCherry-Mis18α (panel **D**), or EGFP-M18BP1 variants (panel **E**) on each centromere. The quantification was performed and is presented in the same way described in the legend of *Figure 2E*. DE, T40D/S110E. (**F**) Western blots of co-immunoprecipitation experiments using GFP-Trap_A beads. HeLa CENP-A-SNAP + EGFP-M18BP1¹⁻¹⁴⁰-P2A-T2A-mCherry-Mis18α or EGFP-M18BP1¹⁻¹⁴⁰ᵀ⁴⁰ᴰ/ˢ¹¹⁰ᴱ-P2A-T2A-mCherry-Mis18α were analyzed.

Mis18α:6His-Mis18β complex contains four MBPs, thus demonstrating that the stoichiometry of the Mis18α:Mis18β complex is 4:2 (*Figure 5C–D*). Another notable feature of the Mis18α:Mis18β complex is an elongated conformation, as suggested by the early elution volume of SEC analysis and the high frictional ratio emerging from the AUC analysis in comparison to that of globular standard proteins (*Erickson, 2009*).

Nardi and colleagues reported that purified full-length His-Mis18α and Strep-Mis18β form a hetero-tetramer when mixed at a 1:1 molar ratio (*Nardi et al., 2016*). Since their experiments were

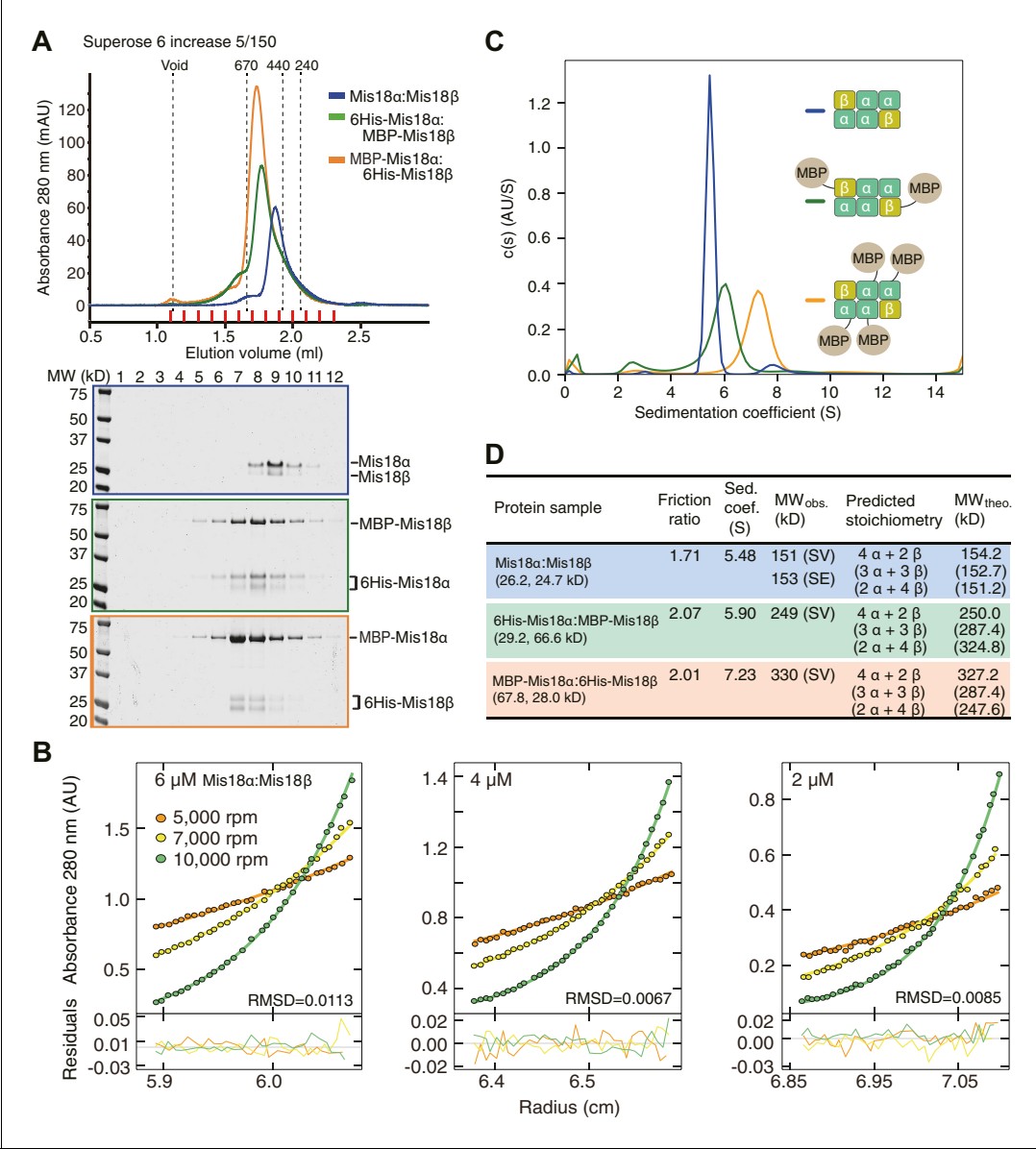

**Figure 5.** Determination of the stoichiometry of Mis18α:Mis18β complex. (**A**) Analytical SEC results of Mis18α:Mis18β (blue), 6His-Mis18α:MBP-Mis18β (green), MBP-Mis18α:6His-Mis18β (orange). The elution volumes of thyroglobulin (670 kD), ferritin (440 kD), and catalase (240 kD) are shown as standards. Red lines indicate fractions collected for Tricine–SDS-PAGE analyses. Gels were stained with CBB. (**B**) Sedimentation equilibrium AUC analysis of Mis18α:Mis18β. Results of data fitting of three datasets obtained at different protein concentrations are shown. Using the global fitting method of SEDPHAT together with four additional datasets (*Figure 5—figure supplement 1A–B*), we obtained the experimental MW of 153 kD. The values of root-mean-square deviation (RMSD) for data fitting are shown. (**C**) Sedimentation velocity AUC results of the same samples used in the analytical SEC experiments (panel A). The best-fit size distributions are shown with the colors indicated in panel A. Data profiles used for curve-fitting analyses are shown in *Figure 5—figure supplement 1C*. (**D**) Summary table of the results obtained from the AUC experiments of panel B and **C**. Sed. coef., sedimentation coefficient; MW_obs., observed molecular weight; MW_theo., theoretical molecular weight; SV, sedimentation velocity; SE, sedimentation equilibrium.

The following figure supplement is available for figure 5:

**Figure supplement 1.** Data profiles for AUC experiments.

carried out with bacterially expressed proteins, whereas we used proteins expressed in insect cells, we asked if the inconsistency of hydrodynamic analyses resulted from differences in the host used for recombinant expression of the Mis18 complex. We therefore co-expressed 6His-Mis18α and MBP-Mis18β in *E. coli* with a bicistronic expression system. After removal of affinity tags, the *E. coli* and insect cell expressed Mis18α:Mis18β complexes showed essentially identical elution profiles in analytical SEC experiments at two different concentrations (*Figure 6A*). The two complexes also behaved essentially identically in sedimentation velocity AUC experiments (*Figure 6B–C* and *Figure 6—figure supplement 1*). These results are in agreement with our contention that the Mis18 complex is hexameric. Untagged Mis18α and Mis18β were insoluble when expressed in isolation in *E. coli* (unpublished data), but tagging them with 6His-MBP delivered soluble products. 6His-MBP-Mis18β appeared monodisperse in analytical SEC experiments, whereas 6His-MBP-Mis18α appeared polydisperse, despite good purity (*Figure 6—figure supplement 2*). Removing the 6His-MBP-tag without prior mixing of 6His-MBP-Mis18α and 6His-MBP-Mis18β caused precipitation of both Mis18α and Mis18β, indicative of poor stability of the individual untagged subunits. Tag cleavage after subunit mixing at 1:1 molar ratio yielded (in addition to a precipitate that was removed by centrifugation) Mis18α:Mis18β complex that eluted from an analytical SEC column at the same volume (~1.45 ml) as the Mis18α:Mis18β complex obtained by co-expression in *E. coli* or insect cells (*Figure 6—figure supplement 2*, compare with *Figure 6A*). Mixing 6His-MBP-Mis18α and 6His-MBP-Mis18β at 2:1 molar ratio and cleaving off the tags reduced precipitation and increased the final yield of soluble Mis18α:Mis18β complex, without changing the elution volume of the complex (*Figure 6—figure supplement 2A*). Finally, we repeated these SEC experiments using the detergent-containing SEC buffer reported by Nardi and colleagues (*Nardi et al., 2016*), again without observing significant changes in the elution profile of the Mis18α:Mis18β complex (*Figure 6—figure supplement 2B*).

## Trimers of C-terminal helices and dimers of Yippee domains mediate hexamer formation

To dissect the mechanism of Mis18 oligomerization, we generated a series of co-expression constructs of Mis18α:Mis18β variants and characterized the purified protein complexes using analytical SEC and sedimentation velocity AUC (*Figure 6A–C*). Both Mis18α$^{1-191}$:Mis18β$^{1-189}$ (N-terminal tail + Yippee domain) and Mis18α$^{78-191}$:Mis18β$^{73-189}$ (Yippee domain) form hetero-dimers (*Figure 6A–C*), in agreement with a recent report (*Subramanian et al., 2016*). On the other hand, the three complexes encompassing the C-terminal helices of Mis18α and Mis18β (6His-Mis18α$^{192-233}$-mCherry: MBP-Mis18β$^{190-229}$, Mis18α$^{192-233}$-mCherry:Mis18β$^{190-229}$, 6His-Mis18α$^{192-233}$:MBP-Mis18β$^{190-229}$) had molecular masses consistent with the presence of two Mis18α and one Mis18β subunit (*Figure 6A–C*), indicating that this is the region of the complex that established the 2:1 ratio of the Mis18α and Mis18β subunits. When considering previous evidence that the Mis18α Yippee domain can also homo-dimerize (*Subramanian et al., 2016*), we can now propose a schematic model for the Mis18α:Mis18β complex (*Figure 6F*) in which the trimeric C-terminal helices promote the assembly of three dimeric Yippee domain interfaces, two of which being α:β and one being α:α.

To identify the binding site for M18BP1$^{1-140}$ on the Mis18α:Mis18β complex, we immobilized MBP-M18BP1$^{1-140}$ on an amylose resin and used it as an affinity reagent to pull-down the Mis18α: Mis18β variants described previously in this section. We found that MBP-M18BP1$^{1-140}$ bound the Mis18α:Mis18β Yippee domain dimer but not the C-terminal helices (*Figure 6D*). Binding was specific for the Mis18α:Mis18β heterodimeric arrangement of Yippee domains, because neither 6His-Mis18α$^{78-191}$ nor 6His-Mis18β$^{73-189}$ bound M18BP1$^{1-140}$ in isolation (*Figure 6E*).

## Mis18α:Mis18β-mediated dimerization of M18BP1 strengthens centromere association

We were able to reconstitute the core of the Mis18 complex by combining Mis18α:Mis18β with M18BP1$^{1–140}$-MBP or M18BP1$^{1–228}$-MBP. The purified complexes appeared monodisperse in SEC profiles (*Figure 7A*), and we obtained an estimate of their MW by sedimentation velocity AUC (*Figure 7B–C*). In line with the results in the previous section, Mis18α:Mis18β:M18BP1$^{1–140}$-MBP and Mis18α:Mis18β:M18BP1$^{1–228}$-MBP have the molecular masses expected for a single Mis18α:Mis18β hexamer plus two M18BP1$^{1–140}$-MBP or M18BP1$^{1–228}$-MBP moieties (*Figure 7C*). The distribution

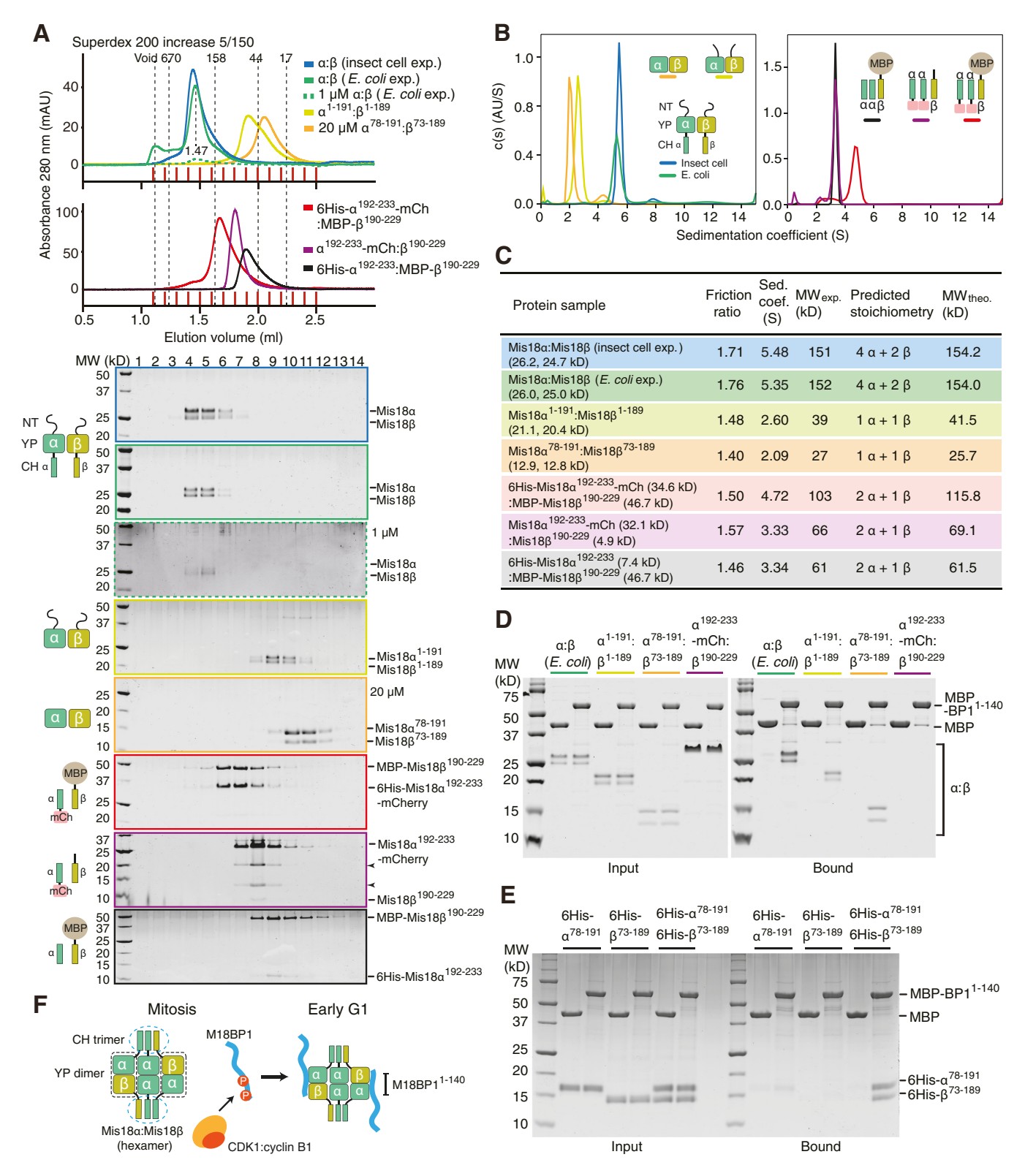

**Figure 6.** Assembly mechanism of Mis18α:Mis18β-hexamer. (**A**) Analytical SEC results of insect-cell-expressed Mis18α:Mis18β (blue), *E. coli*-expressed Mis18α:Mis18β (green), Mis18α$^{1-191}$:Mis18β$^{1-189}$ (yellow-green), Mis18α$^{78-191}$:Mis18β$^{73-189}$ (orange), 6His-Mis18α$^{192-233}$-mCherry:Mis18β$^{190-229}$ (red), Mis18α$^{192-233}$-mCherry:Mis18β$^{190-229}$ (purple), 6His-Mis18α$^{192-233}$:Mis18β$^{190-229}$-MBP (black). SEC experiments were carried out with 10 µM protein (loading concentration), unless indicated in the figure. The elution volumes of thyroglobulin (670 kD), aldolase (158 kD), ovalbumin (44 kD), and

*Figure 6 continued on next page*

*Figure 6 continued*

myoglobin (17 kD) are shown as standards. Red lines indicate fractions collected for Tricine–SDS-PAGE analyses. Gels were stained with CBB, except the gel for 1 µM Mis18α:Mis18β (dashed green line), which was stained with SYPRO Ruby. Left-pointing arrowheads indicate degradation products of Mis18α$^{192-233}$-mCherry. NT, N-terminal tail; YP, Yippee domain; CH, C-terminal helix; mCh, mCherry. (B) Sedimentation velocity AUC results of the same samples used in the analytical SEC experiments (panel A). The best-fit size distributions are shown with the colors indicated in panel A. Data profiles used for curve-fitting analyses are shown in *Figure 6—figure supplement 1*. (C) Summary table of the results obtained from the AUC experiments of panel B. Sed. coef., sedimentation coefficient; MW$_{obs.}$, observed molecular weight; MW$_{theo.}$, theoretical molecular weight. (D, E) Amylose-resin pull-down assays to examine the interaction of Mis18α:Mis18β variants with M18BP1$^{1-140}$. Incubation of amylose beads and proteins (at 5 µM concentration) were performed using a binding buffer containing 30 mM HEPES pH 7.5, 100 mM NaCl, and 1 mM TCEP. Gels were stained with CBB. (F) Hypothetical assembly mechanism of the Mis18α:Mis18β-hexamer in mitosis and the octameric Mis18 complex in the early G1 phase.

The following figure supplements are available for figure 6:

**Figure supplement 1.** Data profiles for AUC experiments.

**Figure supplement 2.** Generation of Mis18α:Mis18β complex by mixing 6His-MBP-Mis18α and 6His-MBP-Mis18β.

plots of the complexes showed single prominent peaks, suggesting that these samples contain a single predominant species (*Figure 7B*). We conclude that the core of the human Mis18 complex contains four Mis18α subunits, two Mis18β subunits, and binding sites for two M18BP1 N-terminal regions (*Figure 6F*).

This binding model predicts that M18BP1 can form dimers in HeLa cells by forming a complex with Mis18α:Mis18β. To test this idea, we generated HeLa CENP-A-SNAP cell lines to co-express EGFP-M18BP1$^{1-140}$ and mCherry-M18BP1$^{1-140}$. As negative control, we co-expressed EGFP-M18BP1$^{1-140/T40D/S110E}$ and mCherry-M18BP1$^{1-140/T40D/S110E}$. We performed co-IP experiments with anti-GFP antibody conjugated beads using lysates from these HeLa cell lines previously treated with the CDK1 inhibitor RO-3306. EGFP-M18BP1$^{1-140}$ co-immunoprecipitated with mCherry-M18BP1$^{1-140}$ and Mis18β, while EGFP-M18BP1$^{1-140/T40D/S110E}$ failed to co-immunoprecipitate with either, indicating that dimerization of M18BP1$^{1-140}$ requires binding to Mis18α:Mis18β in HeLa cells (*Figure 7D*).

We asked if the role of the Mis18α:Mis18β complex as a trigger of M18BP1 dimerization could be rescued by fusing M18BP1 to GST, which is known to form homodimers (*Kaplan et al., 1997*; *Lim et al., 1994*). We therefore generated HeLa CENP-A-SNAP cell lines co-expressing GST-EGFP-M18BP1$^{1-140}$ and GST-mCherry-M18BP1$^{1-140}$, or GST-EGFP-M18BP1$^{1-140/T40D/S110E}$ and GST-mCherry-M18BP1$^{1-140/T40D/S110E}$. Co-IP experiments showed that GST did not interfere with the interaction of M18BP1$^{1-140}$ with Mis18β (*Figure 7D*). As predicted, however, GST rescued the dimerization defect caused by phosphomimetic mutations that prevent the binding of M18BP1 to the Mis18α:Mis18β complex.

We hypothesized that a physiological meaning of the Mis18-mediated dimerization of M18BP1 is to strengthen the affinity of the Mis18α:Mis18β complex for centromeres. To test this hypothesis, we asked if forced dimerization with GST rescued the centromere localization defect of M18BP1 mutants rendering it incapable of binding the Mis18 complex. Toward this end, we generated HeLa CENP-A-SNAP cell lines expressing either GST-EGFP-M18BP1$^{1-1132}$ or GST-EGFP-M18BP1$^{141-1132}$ (*Figure 8*), which are expected to dimerize even without binding Mis18α:Mis18β. GST-EGFP-M18BP1$^{1-1132}$ showed clear localization at centromeres and its expression rescued CENP-A-SNAP loading and Mis18α localization in cells depleted of endogenous M18BP1, indicating that the M18BP1 construct is functional (*Figure 8A and C–E*). GST-EGFP-M18BP1$^{141-1132}$ partially rescued the centromere localization defect of EGFP-M18BP1$^{141-1132}$, with a distribution of GST-EGFP-M18BP1$^{141-1132}$ fluorescence at centromeres that appeared to be significantly stronger and more focused in comparison to that of EGFP-M18BP1$^{141-1132}$ (*Figures 2D*, *8B and E*, *Figure 8—figure supplement 1*). In cells depleted of endogenous M18BP1, GST-EGFP-M18BP1$^{141-1132}$ failed to recruit mCherry-Mis18α to centromeres and did not rescue the defect of CENP-A loading, suggesting that in addition to promoting dimerization and an interaction with centromeres, a physical interaction of M18BP1 with Mis18α:Mis18β is required for CENP-A loading (*Figure 8B–E*).

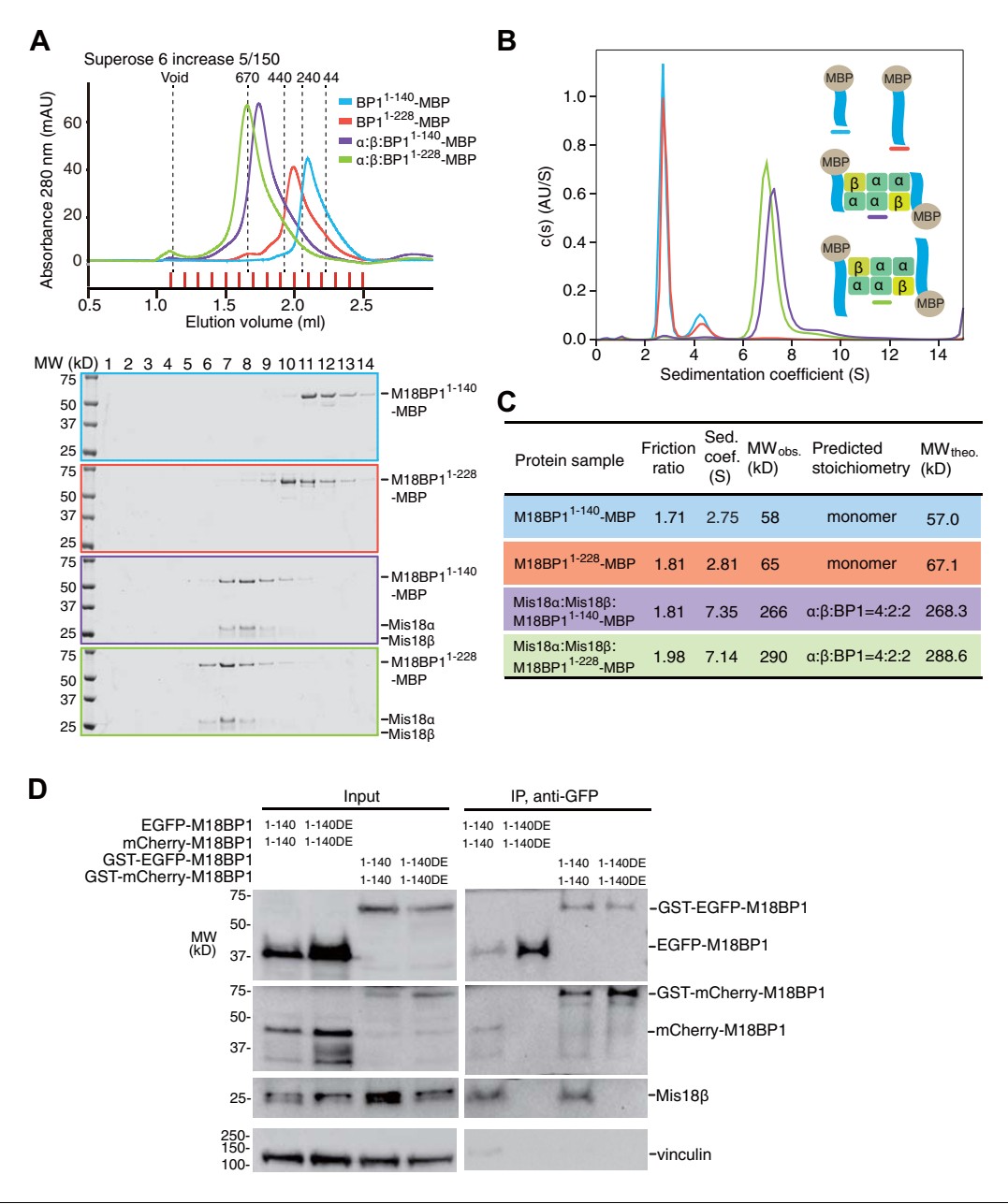

**Figure 7.** Mis18α:Mis18β-hexamer mediates dimerization of M18BP1. (**A**) Analytical SEC results of M18BP1$^{1-140}$-MBP (cyan), M18BP1$^{1-228}$-MBP (red), Mis18α:Mis18β:M18BP1$^{1-140}$-MBP (purple), Mis18α:Mis18β:M18BP1$^{1-228}$-MBP (green). The elution volumes of thyroglobulin (670 kD), ferritin (440 kD), catalase (240 kD) and ovalbumin (44 kD) are shown as standards. Red lines indicate fractions collected for Tricine–SDS-PAGE analyses. Gels were stained with CBB. (**B**) Sedimentation velocity AUC results of the same samples used in the analytical SEC experiments (panel **A**). The best-fit size distributions are shown with the colors indicated in panel **A**. Data profiles used for curve-fitting analyses are shown in *Figure 7—figure supplement 1*. (**C**) Summary table of the results obtained from the AUC experiments of panel **B**. Sed. coef., sedimentation coefficient; MW$_{obs.}$, observed molecular weight; MW$_{theo.}$, theoretical molecular weight. (**D**) Western blot results of co-immunoprecipitation experiments using GFP-Trap_A beads. HeLa CENP-A-SNAP + EGFP-M18BP1$^{1-140}$-P2A-T2A-mCherry-M18BP1$^{1-140}$, EGFP-M18BP1$^{1-140/T40D/S110E}$-P2A-T2A-mCherry-M18BP1$^{1-140/T40D/S110E}$, GST-EGFP-M18BP1$^{1-140}$-P2A-T2A-GST-mCherry-M18BP1$^{1-140}$, or GST-EGFP-M18BP1$^{1-140/T40D/S110E}$-P2A-T2A-GST-mCherry-M18BP1$^{1-140/T40D/S110E}$ were analyzed.

The following figure supplement is available for figure 7:

**Figure supplement 1.** Data profiles for AUC experiments.

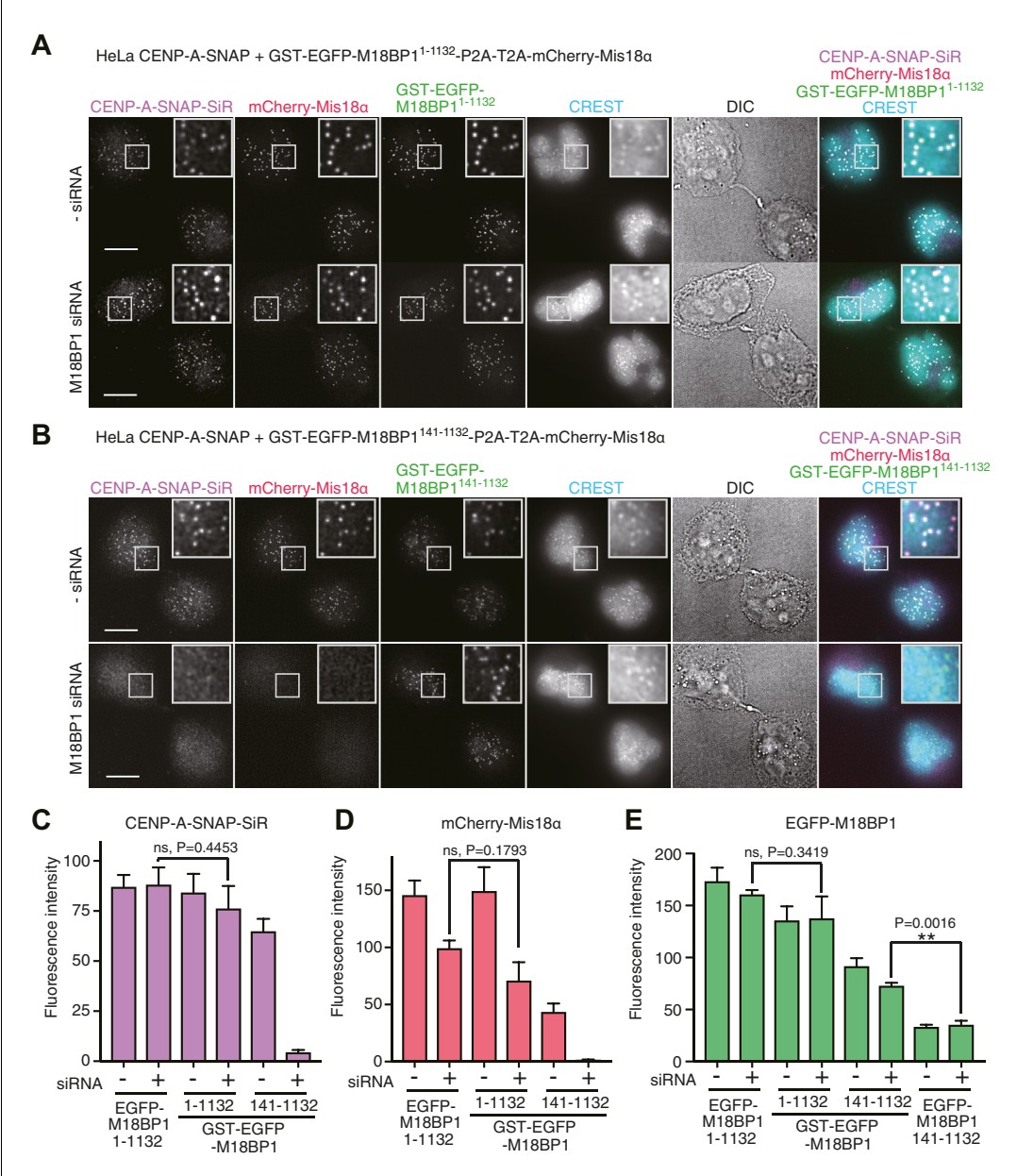

**Figure 8.** Oligomerization of Mis18 complex promotes robust centromere localization. (A, B) Representative images showing the fluorescence of CENP-A-SNAP-SiR, mCherry-Mis18α and GST-EGFP-M18BP1 in the fixed HeLa cells treated as described in *Figure 2A*. Centromeres were visualized with CREST sera. Scale bars represent 10 μm. (C–E) Quantification of the fluorescence intensity of CENP-A-SNAP-SiR (panel C), mCherry-Mis18α (panel D), or EGFP-M18BP1 variants (panel E) on each centromere. The quantification was performed and is presented in the same way described in the legend of *Figure 2*. P-Values are indicated on the graph (Student's t test).

The following figure supplement is available for figure 8:

**Figure supplement 1.** GST-EGFP-M18BP1^141-1132 localizes more specifically to centromeres than EGFP-M18BP1^141-1132.

## Discussion

The evolutionarily conserved Mis18 complex is an essential component of the machinery that loads new CENP-A onto centromeres, but the molecular basis of its assembly had remained unclear. Here, we have reported that the first 140 residues of M18BP1 contain binding motifs that are necessary

and sufficient for a strong interaction with Mis18α:Mis18β in vitro. This result extends very recent work from two other groups, which, by performing ectopic localization experiments with LacO or tetO arrays, identified a Mis18α:Mis18β binding region in longer N-terminal fragments (1–383 or 1–375) of M18BP1 (*Ohzeki et al., 2016*; *Stellfox et al., 2016*).

Previous studies suggested that CDK1 activity regulates CENP-A incorporation into centromeres (*McKinley and Cheeseman, 2014*; *Silva et al., 2012*) and that part of this regulation is through phosphorylation of M18BP1, which prevents complex formation with Mis18α:Mis18β (*McKinley and Cheeseman, 2014*). However, in the absence of a molecular basis for the interaction of M18BP1 with Mis18α:Mis18β, the details of this regulation had remained unclear. We show that each of the two Mis18α:Mis18β binding regions of M18BP1$^{1-140}$ (1-60 and 61–140) contains a highly conserved CDK-phosphorylation motif. Using a series of purified M18BP1 proteins with non-phosphorylatable and phosphomimetic mutations, we demonstrated that Thr40 and Ser110 are targets of the CDK1: Cyclin B complex. When introduced in full length M18BP1, the phosphomimetic mutations T40D and S110E reduced recruitment of M18BP1, prevented recruitment of Mis18α to centromeres, and interfered with new CENP-A deposition. These defects are very similar to the defects observed upon expression of EGFP-M18BP1$^{141-1132}$ in cells depleted of endogenous M18BP1, and point to M18BP1$^{1-140}$ as a controlled interaction module for complex formation with Mis18α:Mis18β.

We used biochemical reconstitution to gain further insights into the organization of the Mis18 complex. The single-peak elution profile from analytical SEC and the single-peak sedimentation-coefficient-distribution profile from AUC of the Mis18α:Mis18β:M18BP1$^{1-140}$-MBP complex indicate that the reconstituted samples are stable and homogeneous. We used these samples to determine the stoichiometry of human Mis18α:Mis18β and of the Mis18 core complex by *Stankovic et al., 2016* AUC. The Mis18α:Mis18β complex is a hexamer containing four Mis18α and two Mis18β subunits, and it binds two M18BP1$^{1-140}$ subunits. The functional importance of oligomerization of Mis18α and Mis18β for CENP-A loading has been highlighted in two additional recent studies (*Nardi et al., 2016*; *Subramanian et al., 2016*). The single fission yeast ortholog of Mis18 uses both the Yippee and the C-terminal coiled-coil domains to form homo tetramers (*Subramanian et al., 2016*). Full-length human Mis18α and Mis18β, on the other hand, have been reported to oligomerize via their C-terminal coiled-coil domains to form 2:2 heterotetramer (*Nardi et al., 2016*). Our results are inconsistent with this recent report. *Nardi et al. (2016)* derived the molecular mass of the Mis18α: Mis18β complex through the Siegel-Monty equation (*Siegel and Monty, 1966*) with estimates of the Stokes radius (from SEC experiments) and of the sedimentation coefficient (from glycerol-gradient ultracentrifugation experiments) derived from Western blot analysis of SDS-PAGE, an approach that we deem more error prone in comparison to our combined sedimentation velocity and sedimentation equilibrium AUC analyses. We show that recombinant Mis18α:Mis18β complexes produced in insect cells and *E coli* have essentially identical hydrodynamic properties in different buffers and concentrations. Thus, the hexamer is the prevalent form of the Mis18α:Mis18β complex.

Our biochemical assays show that the Yippee domains of Mis18α and Mis18β are both required for achieving a strong interaction with M18BP1$^{1-140}$. This suggests that the stoichiometry of Mis18β in the Mis18 complex limits the number of bound M18BP1, because two Mis18α:Mis18β Yippee domain hetero-dimers can be formed in one hexamer, and each can bind a copy of M18BP1$^{1-140}$. Our results are consistent with evidence that M18BP1 interacts with the Mis18α:Mis18β complex only when both Mis18α and Mis18β subunits are co-expressed in a yeast-three-hybrid assay (*Fujita et al., 2007*). The 4:2:2 stoichiometry of the Mis18 complex likely provides a base to understand the assembly of the CENP-A loading machinery, and further structural analysis will have to examine the details of this unique octameric complex.

Overexpression of EGFP-M18BP1$^{1-140}$ in HeLa cells had a strong dominant-negative effect on centromere localization of Mis18α:Mis18β and on CENP-A loading, suggesting that EGFP-M18BP1$^{1-140}$ binds to Mis18α:Mis18β in HeLa cells and prevents its interaction with endogenous M18BP1. This result also indicates that the complex of EGFP-M18BP1$^{1-140}$ with Mis18α:Mis18β cannot localize to centromeres. Indeed, we observed that M18BP1$^{141-1132}$, which is unable to bind with Mis18α: Mis18β, localized to centromeres, although weakly, arguing that M18BP1$^{141-1132}$ contains interaction modules for autonomous localization to centromeres. Mouse M18BP1 and frog M18BP1 isoforms were shown to interact with the CCAN protein CENP-C (*Dambacher et al., 2012*; *Moree et al., 2011*) through a binding domain located between the SANTA and SANT domains (*Dambacher et al., 2012*; *Stellfox et al., 2016*). We demonstrated that forced dimerization of

M18BP1[141-1132] using GST-tag increased its localization to centromeres in HeLa cells, supporting the hypothesis that Mis18$\alpha$:Mis18$\beta$-mediated dimerization of M18BP1 is required to strengthen the interaction between the Mis18 complex and centromeric recruiters. However, GST-mediated dimerization did not rescue centromeric localization of EGFP-M18BP1[141–1132] to the levels of full length EGFP-M18BP1, indicating that the interaction with Mis18$\alpha$:Mis18$\beta$ contributes to the localization mechanism even beyond dimerization. A physical interaction of Mis18$\beta$ with the CENP-C C-terminal region might account for increased binding affinity (*Stellfox et al., 2016*).

Combining our findings with those of recent studies (*Nardi et al., 2016*; *Ohzeki et al., 2016*; *Stellfox et al., 2016*; *Subramanian et al., 2016*), we conclude that full assembly of the Mis18 complex is important for its function. Our analysis demonstrates that the phosphorylation of Thr40 and Ser110 of M18BP1 by CDK1 prevents assembly of the Mis18 complex, thus describing a crucial mechanism to temporally restrict CENP-A loading to early G1 phase, when the levels of Cyclin B rapidly decline after their mitotic peak. In conclusion, the available evidence depicts a complex binding mechanism for the centromere localization of the Mis18 complex, with multiple binding interfaces and a complex regulation by mitotic kinases. Here, we have dissected the regulation of a crucial element of this interface.

# Materials and methods

## Plasmids

Plasmid pETDuet-8His was generated from pETDuet-1 (Novagen) by adding the coding DNA sequence (CDS) of 8His-tag immediately after the *Xho*I site using an modified inverse PCR method (*Erster and Liscovitch, 2010*). Plasmid pETDuet-MBP-8His was generated by inserting the CDS of MBP (*E. coli* malE Lys27–Lys396) followed by a tobacco etch virus (TEV) protease cleavage-site between *Nco*I and *Xho*I sites of pETDuet-8His. Restriction sites of *Bgl*II and *Nco*I in the CDS of MBP were removed and two mutations (A312V/I317V) (*Walker et al., 2010*) were introduced into MBP with a site-directed mutagenesis PCR method (*Sawano and Miyawaki, 2000*). Codon-optimized cDNA of human M18BP1 was purchased from GeneArt. The CDSs of M18BP1 fragments were subcloned into pETDuet-MBP-8His plasmid between TEV-protease cleavage site and 8His-tag using *Bam*HI and *Xho*I sites to generate pETDuet-MBP-M18BP1-8His plasmids. Plasmid pGEX6PT-M18BP1[1–1132]-MBP was generated by inserting the CDS of a TEV-protease cleavage site followed by M18BP1 and MBP (modified sequence described above) between *Bam*HI and *Not*I sites of pGEX-6P-1 (GE Healthcare). The original *Bam*HI site of pGEX-6P-1 was removed by ligation with *Bgl*II site while a new *Bam*HI site was introduced after the TEV-protease cleavage site. A *Xho*I site was introduced between M18BP1 and MBP to enable subcloning of M18BP1 fragments using *Bam*HI and *Xho*I sites. Mutations were introduced using the PCR method (*Sawano and Miyawaki, 2000*).

Codon-optimized cDNAs of human Mis18$\alpha$ and Mis18$\beta$ were purchased from GeneArt. Plasmid pLIB-6His-Mis18$\alpha$ (or Mis18$\beta$) was generated by inserting the CDS of 6His-tag followed by a TEV-protease cleavage site and Mis18$\alpha$ (or Mis18$\beta$) between *Bam*HI and *Sal*I sites of pLIB (*Weissmann et al., 2016*). Plasmid pLIB-MBP-Mis18$\alpha$ (or Mis18$\beta$) was generated by inserting the CDS of MBP followed by a TEV-protease cleavage site and Mis18$\alpha$ (or Mis18$\beta$) between *Bam*HI and *Sal*I sites of pLIB. In these constructs, the original *Bam*HI site of pLIB was abolished by ligation with *Bgl*II site while a new *Bam*HI site was introduced after the TEV-protease cleavage site. Plasmid pLIB-Mis18$\beta$ was generated by inserting the CDS of Mis18$\beta$ between *Bam*HI and *Sal*I sites of pLIB. Co-expression plasmids for baculovirus expression (pBIG1e-6His-Mis18$\alpha$:Mis18$\beta$, pBIG1e-6His-Mis18$\alpha$:MBP-Mis18$\beta$, pBIG1e-6His-Mis18$\beta$:MBP-Mis18$\alpha$) were generated by ligating the DNA fragments of two expression cassettes amplified from pLIB plasmids described above with the pBIG1e backbone (*Weissmann et al., 2016*). *E. coli* co-expression plasmids of pETDuet-6His-Mis18$\alpha$-MBP-Mis18$\beta$ variants were generated by inserting the CDS of Mis18$\alpha$ (or its variants) followed by a ribosome-binding site, the CDSs of MBP and Mis18$\beta$ (or its variants) between *Bam*HI and *Xho*I sites of pETDuet-1. TEV-protease cleavage sites were introduced both between 6His and Mis18$\alpha$ and between MBP and Mis18$\beta$ for removal of 6His-tag and MBP-tag during protein purification. Plasmid pETDuet-6His-Mis18$\alpha$[192-233]-mCherry-MBP-Mis18$\beta$[190-229] was generated by inserting the CDS of mCherry into the plasmid pETDuet-6His-Mis18$\alpha$[192-233]-MBP-Mis18$\beta$[190-229] at the C-terminal side of Mis18$\alpha$[192–233]. Plasmids pETDuet-6His-Mis18$\alpha$[78-191] and pETDuet-6His-Mis18$\beta$[73-189] were generated by inserting

the CDS of a TEV-protease cleavage site followed by the CDS of Mis18$\alpha^{78-191}$or Mis18$\beta^{73-189}$ between *Bam*HI and *Xho*I sites of pETDuet-1. plasmids pETDuet-6His-MBP-Mis18$\alpha$ and pETDuet-6His-MBP-Mis18$\beta$ were generated by inserting the CDS of MBP followed by a TEV-protease cleavage site and Mis18$\alpha$ or Mis18$\beta$ between *Bam*HI and *Xho*I sites of pETDuet-1.

Codon-optimized cDNAs of human CDK1 and Cyclin B1 were obtained from GeneArt. The CDSs of CDK1 and Cyclin B1 were individually subcloned in pLIB with N-terminal GST- and 6His-tag, respectively. They were then subcloned in pBIG1a to generate pBIG1a -GST-CDK1:6His-Cyclin-B1.

Plasmid pcDNA5-EGFP-NLS-P2AT2A-mCherry-PTS1 was generated by inserting the CDS of EGFP-NLS-P2AT2A-mCherry-PTS1 between *Bam*HI and *Bcl*I of pcDNA5/FRT/TO (Thermo Fisher Scientific). The original *Bam*HI site of pcDNA5/FRT/TO was removed by ligation with *Bgl*II site, while a new *Bam*HI site was introduced after the CDS of EGFP. The CDS of NLS can be replaced with the CDS of a protein of interest using *Bam*HI and *Xho*I sites. The CDS of peroxisomal targeting signal–1 (PTS1) can be replaced with the CDS of a protein of interest using *Nhe*I and *Xma*I sites. Plasmid pcDNA5-MTS-TagBFP-EGFP-NLS-P2AT2A-mCherry-PTS1 was generated by inserting the CDS of mitochondrial targeting signal (MTS) fused TagBFP (Evrogen) (MTS-TagBFP) followed by P2AT2A to the upstream of the EGFP of pcDNA5-EGFP-NLS-P2AT2A-mCherry-PTS1.

Plasmids pcDNA5-EGFP-M18BP1-P2AT2A-mCherry-Mis18$\alpha$ variants were generated by replacing the CDSs of NLS and PTS1 of pcDNA5-EGFP-NLS-P2AT2A-mCherry-PTS1 with these of M18BP1 variants and Mis18$\alpha$, respectively, using the restriction sites described above. No stop codon was placed between the CDSs of M18BP1 and P2AT2A. Plasmids pcDNA5-GST-EGFP-M18BP1(1-1132)-P2AT2A-mCherry-Mis18$\alpha$ and pcDNA5-GST-EGFP-M18BP1(141-1132)-P2AT2A-mCherry-Mis18$\alpha$ were generated by inserting the CDS of GST between *Hind*III and *Kpn*I sites of pcDNA5-EGFP-M18BP1(1–1132)-P2AT2A-mCherry-Mis18$\alpha$ and pcDNA5-EGFP-M18BP1(141-1132)-P2AT2A-mCherry-Mis18$\alpha$. Plasmids pcDNA5-EGFP-M18BP1(1-140)-P2AT2A-mCherry-M18BP1(1-140) and pcDNA5-EGFP-M18BP1(1-140/T40D/S110E)-P2AT2A-mCherry-M18BP1(1-140/T40D/S110E) were generated by replacing the CDS of Mis18$\alpha$ with the CDS of M18BP1(1–140) or M18BP1(1–140/T40D/S110E).

Plasmid pETDuet-CENP-A-SNAP-HA-PGK-NeoR was generated by ligating four DNA fragments (1 kb CENP-A gene sequence upstream form the stop codon, the CDS of SNAP-3HA, PGK promoter with neomycin resistant gene (NeoR), 1 kb CENP-A gene sequence downstream form the stop codon) and pETDuet backbone treated with *Xba*I and *Xho*I restriction enzymes using Gibson cloning method (*Gibson et al., 2009*). The CDS of SNAP-3HA was amplified from the pSS26m plasmid containing the CDS of CENP-A-SNAP-3HA, which was a gift from L.E.T. Jansen (Instituto Gulbenkian de Ciência, Portugal) (*Jansen et al., 2007*).

Plasmid pX330-CENP-A-sgRNA was generated by inserting annealed oligos (5'–CACCGCGAG TCCCTCCTCAAGGCCC–3', 5'–AAACGGGCCTTGAGGAGGGACTCGC–3') into *Bbs*I sites of pX330 (*Ran et al., 2013*), which was purchased from Addgene.

All constructs were verified by DNA sequencing before being used for protein expression or as the templates for PCR. The list of all plasmids used in this study is presented in *Supplementary file 1*.

## Protein expression

*E. coli* cells of BL21-CodonPlus(DE3)-RIL strain transformed with expression plasmids were cultured in 2xYT media (16 g/L tryptone, 10 g/L yeast extract, 5 g/L NaCl) supplemented with ampicillin and chloramphenicol at 37°C. Protein expression was induced by adding IPTG to the final concentration of 0.2 mM when OD$_{600}$ of the culture reached 0.6 and further incubation at 20°C for 16 hr.

Plasmids of pBIG1 derivatives carrying expression cassettes were recombined to baculoviral genome by Tn7 transposition in *E. coli* DH10EMBacY cells (*Trowitzsch et al., 2010*). Recombinant EMBacY with expression cassettes were extracted from bacteria cells and used to transfect Sf9 cells for baculovirus generation. Initial viruses were obtained after 3 days incubation of EMBacY and Sf9 cells with FuGENE reagent (Promega). Viruses were amplified using Sf9 cells. Litter-scale protein expression was performed by incubation of viruses and Tnao38 cells at 27°C for 4 days with starting density of 1.2 million Tnao38 cells per mL of Sf-900 III SFM media (Thermo Fisher Scientific).

## Protein purification

All purification procedures were performed either on ice or at 4°C. C-terminal MBP-tagged M18BP1 protein samples were purified from *E. coli* cells expressing GST-M18BP1-MBP variants. The cells were suspended in buffer containing 50 mM HEPES pH 7.5, 500 mM NaCl, 1 mM tris(2-carboxyethyl)phosphine hydrochloride (TCEP) and 1 mM PMSF and lysed by sonication. The clear supernatant obtained after centrifugation was incubated with glutathione sepharose resin (GE Healthcare) for ~16 hr. Protein-bound resin was washed with 100 column volumes of buffer A and incubated for ~16 hr with His-tagged TEV protease (His-TEVp) for on-resin cleavage at a stoichiometry of 1/10 with respect to the expected yield of the GST-M18BP1-MBP variant. His-TEVp cut between GST and M18BP1. Eluted M18BP1-MBP variant was separated using Ni resin (cOmplete His-tag purification resin, Roche) from His-TEVp and concentrated to more than 5 mg/ml.

N-terminal MBP-tagged M18BP1 protein samples were purified from *E. coli* cells expressing MBP-M18BP1-8His variants. The cells were suspended in buffer HST300 (30 mM HEPES pH 7.5, 300 mM NaCl, 1 mM TCEP) containing 1 mM PMSF and 10 mM imidazole and lysed by sonication. The clear supernatant obtained after centrifugation was incubated with Ni resin (Roche) for ~16 hr. Protein-bound resin was washed with 100 column volumes of buffer HST300 containing 10 mM imidazole. MBP-M18BP1-8His variants were eluted with 10 column volumes of buffer HST300 containing 400 mM imidazole and concentrated in buffer HST300 to more than 5 mg/ml. The concentration of imidazole were reduced to less than 10 mM by repetition twice of dilution of the concentrated protein sample with 10 times buffer HST300. MBP-8His was purified in the same way as MBP-M18BP1-8His.

Insect-cell-expressed Mis18α:Mis18β complex was purified from Tnao38 cells co-expressing 6His-tagged Mis18α and untagged Mis18β. The cells were suspended in buffer HST300 containing 10 mM imidazole and subjected to Ni-affinity purification using the same procedure as for purification of MBP-M18BP1-8His. Concentrated 6His-Mis18α:Mis18β complex was incubated with His-TEVp for ~16 hr to cleave off the His-tag. Mis18α:Mis18β complex was further purified using Superdex 200 10/300 GL SEC column (GE Healthcare) and concentrated to 5 mg/mL in buffer HST300. *E. coli*-expressed Mis18α:Mis18β variants were purified using the same procedure as for purification of insect-cell-expressed Mis18α:Mis18β complex. MBP-Mis18α:6His-Mis18β and 6His-Mis18α:MBP-Mis18β complexes were also purified from Tnao38 cells using the same procedure as for purification of Mis18α:Mis18β complex without the step of tag-cleavage using His-TEVp.

Purified Mis18α:Mis18β complex (insect cell expression) was incubated with 1.5–2 times of (molar ratio) M18BP1$^{1–140}$-MBP or M18BP1$^{1–228}$-MBP in buffer HST300. The Mis18α:Mis18β:M18BP1$^{N-terminus}$MBP complexes were separated from excess M18BP1$^{1–140}$-MBP or M18BP1$^{1–228}$-MBP using SEC with Superdex 200 10/300 GL SEC column equilibrated in buffer HST300.

GST-CDK1:6His-Cyclin B1 was purified from Tnao38 cells using glutathione Sepharose (GE Healthcare) followed by size exclusion chromatography using HiLoad 16/60 Superdex 200 pg column (GE Healthcare) equilibrated in buffer containing 20 mM HEPES pH 7.5, 200 mM NaCl, 1 mM TCEP, and 5% glycerol.

## Phosphorylation using CDK1:Cyclin B1

Purified MBP-M18BP1-8His variants and MBP-8His were incubated at the concentration of 10 μM with 50 nM GST-CDK1:6His-Cyclin B1 complex in the reaction solution containing 20 mM HEPES pH 7.5, 300 mM NaCl, 1 mM TCEP, 5 mM ATP, and 10 mM MgCl$_2$ at 30°C for 2 hr.

## Pull down assays using amylose resin

Proteins were diluted to 3 μM in 40 μL buffer HST300, unless otherwise noted, and mixed with 20 μL amylose resin (NEB) equilibrated with buffer HST300 or buffer with reduced NaCl concentration of 100 mM. One-third of this mixture was taken as input fraction and the rest two-thirds were incubated at 4°C for 30 min. Amylose-bound proteins were separated from unbound fraction by spinning down the amylose resin and washing with 500 μL buffer HST300 four times. The input and bound fractions were analyzed by Tricine–SDS-PAGE using normal gels and Phos-tag gels. The gels were stained with Coomassie brilliant blue (CBB). Phos-tag acrylamide gels containing 50 μM Phos-tag AAL-107 (NARD institute) were prepared according to the manufacture's protocol and were used to detect the mobility shift caused by phosphorylation.

## Generation of HeLa cell lines

A HeLa CENP-A-SNAP cell line was directly generated from a Flp-In T-REx HeLa cell line generated by Stephen Taylor and colleagues (*Tighe et al., 2008*), which we did not further authenticate. All cell lines are described in *Supplementary file 2*. We tagged the C-terminus of endogenous CENP-A loci with a SNAP-3HA-tag using CRISPR/Cas9-mediated genome engineering following the protocol previously described (*Ran et al., 2013*) with modifications. The Flp-In T-REx HeLa cells were transfected with pX330-CENP-A-sgRNA together with the PCR-amplified rescue DNA fragment from pETDuet-CENP-A-SNAP-HA-PGK-NeoR that contained the CDS of SNAP-3HA-PGK-NeoR with 1 kb CENP-A genomic DNA flanking both sides. The positive clones were selected using G418, and the correct tagging was verified by confirming the centromere labeling of the early G1 cells with SNAP-Cell 647-SiR (NEB).

HeLa CENP-A-SNAP cell lines co-expressing EGFP-M18BP1 variants and mCherry-Mis18α were generated by transfecting the HeLa CENP-A-SNAP cells with pcDNA5-EGFP-M18BP1-P2AT2A-mCherry-Mis18α plasmids and pOG44 plasmid according to the protocol previously described (*Tighe et al., 2004*, *2008*). HeLa cell lines co-expressing EGFP-NLS and mCherry-PTS1 with or without MTS-TagBFP were generated by transfecting the Flp-In T-REx HeLa cells with the pcDNA5-EGFP-NLS-P2AT2A-mCherry-PTS1 derived plasmids using the same protocol described above. Cell lines were regularly tested for mycoplasma contamination and tests found negative.

## CENP-A loading experiment

HeLa cells were placed in wells of 12-well plates in DMEM (PAN Biotech) supplemented with 10% tetracycline-free FBS (Thermo Fisher Scientific, Waltham, MA) and 2 mM l-glutamine (PAN-Biotech) and grown at 37°C in the presence of 5% $CO_2$ for 24 hr. Then, the cells were treated with Lipofectamine RNAiMAX, serum-free OptiMEM (Thermo Fisher Scientific), 10 nM M18BP1 siRNA (5'-GAAG UCUGGUGUUAGGAAAdTdT-3') (*Fujita et al., 2007*) for 48 hr according to the manufacture's protocol, and the control cells were treated with Lipofectamine RNAiMAX and serum-free OptiMEM without siRNA. Doxycycline (Sigma) was added to the culture at a concentration of 50 ng/ml to induce protein expression and was kept in the media until the fixation of cells. Thymidine (1 mM final concentration) was used to arrest cells at S/G1 transition phase. When cells were released from thymidine, existing CENP-A-SNAP proteins were blocked using SNAP-Cell Block (NEB) according to the manufacture's protocol. STLC (5 μM final concentration) was used to arrest cells in prometaphase. The cells arrested in prometaphase were separated from other cells by mitotic-shake-off, released from STLC by extensive wash with the media and placed in wells of 24-well plates containing poly-lysine coated coverslips. Three hours later, the cells in early G1 phase attached on the coverslips and were treated with SNAP-Cell 647-SiR (NEB) to label newly synthesized CENP-A-SNAP according to the manufacture's protocol. Cells were fixed with PBS/PHEM (Pipes, Hepes, EGTA, and $MgCl_2$)-paraformaldehyde 4% followed by permeabilization with PBS/PHEM–Triton X-100 0.5% and immunostaining. The following antibodies were used for immunostaining: CREST/anticentromere antibodies (human autoimmune serum, Antibodies, Inc.), anti-M18BP1 (Bethyl A302-825A), anti-rabbit Rodamine red–conjugated, anti-human DyLight 405–conjugated secondary antibodies were purchased from Jackson ImmunoResearch Laboratories, Inc. Coverslips were mounted with Mowiol mounting media (EMD Millipore) and imaged using a 60× oil immersion objective lens on a DeltaVision deconvolution microscope. Quantification of centromere signals were performed using the software Fiji with a script from a previous study (*Bodor et al., 2012*) with modifications for semi-automated processing. Briefly, average projections were made from z-stacks of recorded images. Centromere spots were chosen based on the parameters of shape, size, and intensity using the images obtained with CREST-staining, and their positions were recorded. The mean intensity value of adjacent pixels of a centromere spot was subtracted as background intensity from the mean intensity value of the centromere spot. Statistical analysis of the quantified intensity was performed and the plots were generated with the software Prism 7 (GraphPad Software Inc.).

## Co-immunoprecipitation experiment

HeLa cells were grown in DMEM supplemented with 10% tetracycline-free fetal bovine serum, 1% L-glutamine and 1% Penicillin-Streptomycin at 37°C in a 5% $CO_2$ atmosphere. Cells were treated with 50 ng/mL doxycycline (Sigma) for 18 hr and then treated with both 50 ng/mL doxycycline and 9

µM CDK1 inhibitor RO-3306 (Merck) for additional 6 hr before harvest. Cells were washed twice with PBS, resuspended in lysis buffer containing 75 mM HEPES pH 7.5, 1.5 mM EGTA, 1.5 mM MgCl$_2$, 150 mM NaCl, 10% glycerol, 0.1% NP-40, 1 mM PMSF and Protease Inhibitor Mix HP Plus (SERVA), lysed using Bioruptor Plus sonication device (Diagenode) and centrifuged at 16,000 x g for 30 min at 4°C. The supernatant containing 2 mg protein was incubated with 7.5 µl GFP-Trap_A beads (Chromotek) for 2 hr at 4°C. The beads were washed thrice with the lysis buffer containing 300 mM NaCl instead of 150 mM NaCl. Proteins were eluted by adding SDS sample buffer and were analyzed using Tricine-SDS-PAGE and Western blot analysis. EGFP-M18BP1, mCherry-M18BP1, mCherry-Mis18$\alpha$, Mis18$\beta$ and vinculin were detected using following antibodies: anti-GFP (Abcam, AB6556), anti-mCherry (Novus, NBP1-96752), anti-Mis18$\beta$ (Atlas, HPA052271), anti-vinculin (Sigma, V9131) anti-mouse-HRP (Amersham, NXA931-1ML) and anti-rabbit-HRP (Amersham, NXA934-1ML).

## Analytical size-exclusion chromatography

Size-exclusion chromatography (SEC) experiments were performed on calibrated Superose 6 increase 5/150 GL column or Superose 200 increase 5/150 GL column (GE Healthcare). Purified protein samples were applied to the column at 10 µM, unless otherwise noted, and eluted under isocratic condition at 4°C in buffer HST300 at a flow rate of 0.1 ml/min. Fractions were collected and analyzed by Tricine–SDS-PAGE and the gels were stained with CBB or SYPRO Ruby (Thermo Fisher Scientific).

## Analytical ultracentrifugation

Sedimentation velocity AUC was performed at 42,000 rpm at 20°C in a Beckman XL-A ultracentrifuge. Purified protein samples were diluted to 0.3–0.5 mg/mL in buffer HST300 and loaded into standard double-sector centerpieces. The cells were scanned at 280 nm every minute and 300 scans were recorded for every sample. Data were analyzed using the program SEDFIT (*Schuck, 2000*) with the model of continuous *c(s)* distribution. The partial specific volumes of the proteins, buffer density, and buffer viscosity were estimated using the program SEDNTERP. Data figures were generated using the program GUSSI.

Sedimentation equilibrium AUC was performed at 20°C using standard Epon six-channel centerpieces. Purified Mis18$\alpha$:Mis18$\beta$ complex was diluted to five different concentrations (6, 4, 2, 1, and 0.5 µM, assuming the MW of the complex is 154 kD) and centrifuged at 5000, 7000, and 10000 rpm until samples in the cells reached sedimentation equilibrium. Protein sedimentation was recorded at 280 and/or 230 nm. Data were processed using the program SEDFIT and analyzed using the program SEDPHAT (www.analyticalultracentrifugation.com).

## Acknowledgements

We thank Lars ET Jansen for sharing reagents. We thank the members of the Musacchio laboratory, especially Stefano Maffini, Charlotte Smith, and Katharina Overlack for helpful advice and discussions. AM acknowledges funding by the European Union's seventh Framework Program ERC advanced grant agreement RECEPIANCE and the DFG's Collaborative Research Centre (CRC) 1093. DP and PS gratefully acknowledge support from the Alexander von Humboldt Foundation through Humboldt Research Fellowships.

## Additional information

### Competing interests

AM: Reviewing editor, *eLife*. The other authors declare that no competing interests exist.

### Funding

| Funder | Grant reference number | Author |
| --- | --- | --- |
| European Research Council | AdG 669686 RECEPIANCE | Andrea Musacchio |
| Deutsche Forschungsgemeinschaft | Collaborative Research Centre (CRC) 1093 | Andrea Musacchio |

| Alexander von Humboldt-Stif-tung | Postdoctoral fellowship | Dongqing Pan Priyanka Singh |

The funders had no role in study design, data collection and interpretation, or the decision to submit the work for publication.

## Author contributions
DP, Conception and design, Acquisition of data (generated constructs and purified proteins, performed pull-down assays and analytical SEC experiments, generated HeLa CENP-A-SNAP cell line, generated HeLa cell lines and performed CENP-A loading experiments, and performed AUC experiments), Analysis and interpretation of data, Drafting or revising the article; KK, Acquisition of data (generated HeLa CENP-A-SNAP cell line, generated HeLa cell lines and performed CENP-A loading experiments), Analysis and interpretation of data; AP, Acquisition of data (performed AUC experiments), Analysis and interpretation of data; AT, Acquisition of data (generated HeLa cell lines and performed CENP-A loading experiments); KW, Acquisition of data (generated HeLa cell lines, performed CENP-A loading experiments and performed co-IP experiments), Analysis and interpretation of data; PS, Contributed unpublished essential data or reagents (generated constructs and purified proteins); AR, Acquisition of data (generated HeLa CENP-A-SNAP cell line), Analysis and interpretation of data; AWB, Analysis and interpretation of data (generated HeLa CENP-A-SNAP cell line); AM, Conception and design, Analysis and interpretation of data, Drafting or revising the article

## Author ORCIDs
Arnaud Rondelet, http://orcid.org/0000-0001-5816-4137
Alexander W Bird, http://orcid.org/0000-0002-1061-0799
Andrea Musacchio, http://orcid.org/0000-0003-2362-8784

## Additional files

### Supplementary files
• Supplementary file 1. Plasmid vectors used in this study.

• Supplementary file 2. HeLa cell lines used in this study.

• Supplementary file 3. Fluorescence intensity data.

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
