## [Decision Letter]

[Editors’ note: a previous version of this study was rejected after peer review, but the authors submitted for reconsideration. The first decision letter after peer review is shown below.]

Thank you for submitting your work entitled "CDK-regulated oligomerization of M18BP1 on a Mis18 hexamer is necessary for CENP-A loading" for consideration by *eLife*. Your article has been favorably evaluated by Jessica Tyler (Senior Editor) and three reviewers, one of whom, Asifa Akhtar, is a member of our Board of Reviewing Editors.

The reviewers appreciated that this manuscript presents a thorough biochemical characterization of the Mis18 heterohexamer and that specific M18BP1 phosphosites regulate Mis18α/β binding. Although these contributions are worthwhile, the fact that M18BP1 phosphorylation regulates Mis18α/β binding has been reported previously which impinges on the overall novelty of the manuscript. Moreover, since very different experimental protocols for reconstitutions are used between this and an earlier study, it remains possible that the complex may exist in different stoichiometry, this issue still remains open for discussion at this stage. Overall, the reviewers were of the opinion that the manuscript represents a modest advance in the field at this stage and discussed that the manuscript could perhaps be improved either by: (1) firmly establishing the biological role of the hexamer vs. a tetramer and by providing a better explanation for the difference between their results and the previous literature, or (2) from better delineating the role of the dimerization of M18BP1 via the Mis18 complex.

Our decision has been reached after consultation between the reviewers. Based on these discussions and the individual reviews below, we regret to inform you that your work will not be considered further for publication in *eLife*.

Reviewer #1:

Overall, this is a very thorough study showing the dependency of Mis18α/Mis18β localisation to the centromeres and the M18BP1 phosphorylation by CDK1/CycB. Their identification of the phosphorylation sites and corresponding mutations for those sites clearly indicates a functional relevance of this modification on M18BP1 and its interaction with Mis18α/Mis18β. Furthermore, they have validated their 2A-peptide dependent assay (Figure 2—figure supplement 2) carefully showing they have a robust and reliable method for their assays.

As they were addressing the minimally required fraction of the M18BP1 they discovered that 1-140aa was sufficient for the Mis18α/Mis18β interaction but this fraction of the protein was showing a dominant negative effect on the endogenous M18BP1 however there is not any assay showing that this could be by direct binding to the endogenous protein or not. Furthermore, the phoshomimics of the same fragment (1-140/T40D/S110E) did not show any dominant negative effect, which the authors explained by stating that the Mis18α/Mis18β interaction is lost. There could be a more convincing proof if the authors performed an IP or pulldown for these two aspects.

They claim that the dimerisation of M18BP1 strengthens Mis18α/b centromere localisation with the GST tagged M18BP1 (Figure 6). This is done by comparing to the earlier assay (Figure 2) however the fluorescence signal should not be quantified between the two independent cell lines unless they can be imaged side-by-side or no conclusion could be drawn from the signal becoming "sharper" (e.g. One explanation could be that the GST tag affects the stability of the proteins).

*Reviewer #2:*

Centromeres are epigenetically specified by the H3 variant CENP-A. In this manuscript, Pan et al., report a series of elegant experiments that provide new insight into the process of CENP-A assembly focusing on the Mis18 complex a set of proteins essential for delivering new CENP-A to centromeres. The authors present a set of biochemical experiments defining the domain of M18BP1 that binds to the Mis18α and β subcomplexes. They show this interaction is blocked by Cdk mediated phosphorylation and phosophomemetic mutations cause a defect in CENP-A assembly in cells.

Purification of the Mis18α/β subcomplex shows a multimeric complex which the authors find to be a hexamere consisting of 4 α and 2 β molecules and that this complex can bind to 2 M18BP1 N terminal fragments.

While this study overlaps with recent work from e.g. the Foltz lab they provide interesting new insights particularly with regard to Cdk regulation which is valuable to the field.

There are however some specific issues regarding the interpretation of some of the data that need to be resolved:

1) The title and text speak of oligomerization of M18BP1. I don´t see any evidence for this. The authors show that the Mis18α/β subcomplex can bind two molecules of an N-terminal M18BP1 fragment. This does not in itself indicate oligomerization which implies a multimerization (i.e. more than two). Using the term "Mis18α/β-induced dimerization of M18BP1" seems to me more appropriate.

2) Subsection “M18BP1^1–140^ contains two sequential binding regions for Mis18α:Mis18β”:

By identifying two consecutive fragments on M18BP1 that both retain the ability to interact with the Mis18α/β proteins does not necessarily define "sequential binding regions" or a "bipartite" (first paragraph) binding site exists as interpreted by the authors. Because the two fragments are adjacent to one another these results are consistent with a single continuous binding interface that is artificially broken into two, each retaining some binding activity. Calling this a bipartite binding site is misleading and should not be described as such.

3) Mis18αβ stoichiometry:

This manuscript presents biophysical data on the stoichiometry of the Mis18α/β interpreted to be 4xalpha with 2x beta subunits. As indicated, this is inconsistent with a previous report. The authors explain this discrepancy by the fact that they co-express the α and β subunit while in the Nardi et al. study (Nardi et al., 2016), α and β are expressed and purified separately and then mixed in a 1:1 ratio.

While this is a possible explanation this does not mean the experiment by the authors is superior or more likely to be correct. Given that Mis18α and β can be expressed separately and able to form a complex indicates that the stoichiometry can be established dynamically from preexisting components and does not need to be established co-translationally.

Further, what is the relative expression level of Mis18α and β in the insect cells used by the authors? Can the skewed α/β ration be a consequence of relative over expression of Mis18α?

At least, the authors should leave open the possibility that the hexamer is one likely conformation but perhaps, depending on the context, other conformations may exist.

4) Figure 2, Figure 4, Figure 6, Figure 2—figure supplement 1:

Why normalize to CREST signals? This is likely to create normalization artifacts as CREST signals appear to be rather poor with high background. Further, CREST stains a region that is broader than the CENP-A domain due to CENP-B staining further complicating normalization. Finally, and importantly, most CREST antibodies recognize, in part, CENP-A which means that the changes in CENP-A signal also potentially affect the normalization signal. CENP-A-SNAP values should simply be background corrected but not normalized.

5) Figure 2, Figure 4 and Figure 6:

Box plots are shown of centromere signal values. The spread in this data represents the biological variance in centromere intensities not the experimental variance. How often are these experiments repeated? In order to make any claim regarding differences and their significance (as in Figure 6) it is meaningless to compare the variance in centromere intensities. When enough centromeres are sampled there will always be a significance difference. What matters is the variance in the experiment itself. i.e. are the means of replicate experiments significantly different? Without this, these quantifications are not informative.

6) Final section:

The argument of oligomerization and its benefit to M18BP1 recruitment is weakly supported by data. Essentially, the only evidence in support of this is the slight increase in M18BP1 localization when it is forced to dimerize by fusion to GST in the absence of its ability to dimerize via Mis18 proteins (by deletion of the N-terminal 140aa).

This experiment is however, poorly controlled. First, as I outlined above the level of significance of the difference in the signal intensities cannot be determined by comparing the variance in centromeres across cells in a signal experiment. Instead, the reproducibility of the difference between experiments should be tested. Secondly, it is not clear from this experiment whether the perceived increase in M18BP1 centromere localization by GST dimerization compensates for the lack of dimerization via the Mis18 proteins. In other words, does GST play a role here that would otherwise be provided by the Mis18 proteins? One way to test this is to fuse GST to full length M18BP1 and test its effect on localization. In this case, if the authors´ hypothesis is correct, then M18BP1 dimerization via M18alpha and beta is functional and further dimerization via GST would have no further contribution.

*Reviewer #3:*

Centromere identity is epigenetically specified by nucleosomes in which H3 has been substituted for the centromere-specific histone H3 variant CENP-A. Faithful, site-specific transmission of CENP-A chromatin from one generation to the next is essential for chromosome segregation and long-term genome stability. This process is carried out by the CENP-A specific chaperone HJURP which is targeted to the proper locus by the Mis18 complex (Mis18α, Mis18β, and M18BP1) immediately following mitosis during early G1. Cell cycle regulated assembly and centromere recruitment of the Mis18 complex is essential for centromere inheritance, and a major focus of recent work in the centromere field has been to understand the basis of this regulation.

Previous studies have shown that Cdk phosphorylation prevents premature recruitment of the Mis18 complex to centromeres during G2 and mitosis (Silva, et al. 2012). This was extended by the demonstration that Cdk phosphorylation of M18BP1 prevents recruitment of Mis18α/β to the N-terminal 383 amino acids of M18BP1 (McKinley and Cheeseman, 2014). In this manuscript, Pan, et al. show that of the residues mutated by McKinley and Cheeseman, two of them in particular, T40 and S110, inhibit the direct interaction of M18BP1 with Mis18α/β when phosphorylated.

M18BP1 must associate with Mis18α/β to target the complex to G1 centromeres thus it is important to understand how the complex is assembled. Previous work using size exclusion chromatography and glycerol gradient sedimentation led to the proposal that the Mis18 complex comprises a Mis18 heterotetramer (2α:2β) formed by their C-terminal coiled-coil domains bound to an unknown number of M18BP1 molecules (Nardi, et al. 2016). In this work, Pan, et al. use sedimentation velocity and equilibrium ultracentrifugation to determine that the Mis18 complex comprises a Mis18 heterohexamer (4α:2β), which in turn can bind 2 copies of the M18BP1 N-terminus. The authors argue that a Mis18 hexamer is the true stoichiometry because the methods used in the previous study are more error-prone. In addition, they suggest that the previous study may have obtained a 2:2 stoichiometry because they combined a 1:1 molar ratio of separately purified Mis18α and Mis18β rather than coexpressing them in eukaryotic cells.

The major scientific contributions in this manuscript are 1) the biochemical characterization of the Mis18 heterohexamer which clarifies previous studies and 2) the specific M18BP1 phosphosites that regulate Mis18α/β binding. Although these contributions are worthwhile and the biochemical studies are well done it is not clear whether this work warrants publication in *eLife* given the relatively modest advance in our understanding of Mis18 complex function in centromere formation.

1) The reason for the discrepancy between the stoichiometry measured by Pan, et al. and Nardi, et al. remains unclear. By size exclusion, Nardi et al. also appear to observe a mono-dispersed species with a migration consistent with their calculated molecular weight. Pan, et al. argue based on their analyses that the Mis18 complex adopts an elongated conformation, leading to its earlier elution in size exclusion chromatography. Therefore, if anything I would expect a molecular weight calculation based on SEC elution of this complex to overestimate rather than underestimate its stoichiometry. In the absence of data supporting the idea that the stoichiometry of the complex is 'titratable,' I also don't find the notion that Nardi et al. observed a tetramer simply because they mixed Mis18α and Mis18β in a 1:1 ratio compelling. If the ratio were off, presumably Nardi et al. would have seen a population at equilibrium containing some hexamer and some monomer/dimer/trimer. The difference in observed species may likely be due to the different expression systems used (bacterial vs. eukaryotic) and resulting differences in protein modifications/folding/etc., but there may also be some interesting biology in the differences. On the other hand, elaborating on the technical differences between the methods to clarify why the Siegel-Monty method is more error-prone would be helpful.

I don't think it's specifically up to the authors in this paper to resolve the reason for this discrepancy, but given that it's not clear whether we're looking at precisely the same complex as in Nardi et al. I think it's important for the authors to provide some additional characterization of the hexamer. Is the coiled-coil domain involved in hexamerization? Do the coiled-coil mutations described by Nardi et al. affect hexamer formation? Is the hexamer comprised of a Mis18α/β tetrameric core mediated by coiled coils with additional Mis18α bound by a different mechanism (e.g., the Yippee domain, which is also used for tetramerization in SpMis18)? The Musacchio lab has made extensive use of cross-linking/mass spectrometry in the past to understand the organization of multi-protein complexes – perhaps this could shed light on differences between the Nardi and Pan Mis18 complexes.

2) Although it's clear that Mis18 complex assembly is required for its recruitment, it is not clear whether the cell cycle timing of its recruitment is strictly determined by regulating the timing of complex assembly or whether regulation of some additional interaction plays a role. Silva, et al. found that mutating 24 putative Cdk sites (including S110 but not T40) in M18BP1 uncoupled its centromere localization from cell cycle regulation, allowing its recruitment in mitosis and G2. Can this be recapitulated by the lone S110A (or T40V/S110A) mutant characterized by Pan, et al.?

---

## [Author Response]

[Editors’ note: the author responses to the first round of peer review follow.]

Reviewer #1:

*[…] As they were addressing the minimally required fraction of the M18BP1 they discovered that 1-140aa was sufficient for the Mis18α/Mis18β interaction but this fraction of the protein was showing a dominant negative effect on the endogenous M18BP1 however there is not any assay showing that this could be by direct binding to the endogenous protein or not. Furthermore, the phoshomimics of the same fragment (1-140/T40D/S110E) did not show any dominant negative effect, which the authors explained by stating that the Mis18α/Mis18β interaction is lost. There could be a more convincing proof if the authors performed an IP or pulldown for these two aspects.*

We thank the reviewer for suggesting the IP experiments. We have now performed co-IP experiments using anti-GFP antibody coupled beads to pull-down either EGFP-M18BP1(1-140) or EGFP-M18BP(1-140/T40D/S110E)

from the HeLa cell lines treated with the CDK1 inhibitor RO-3306. The results, shown in Figure 4, demonstrate that EGFP-M18BP(1-140) interacts with Mis18⟨:Mis18® but EGFP-M18BP1(1-140/T40D/S110E) doesn’t.

*They claim that the dimerisation of M18BP1 strengthens Mis18α/b centromere localisation with the GST tagged M18BP1 (Figure 6). This is done by comparing to the earlier assay (Figure 2) however the fluorescence signal should not be quantified between the two independent cell lines unless they can be imaged side-by-side or no conclusion could be drawn from the signal becoming "sharper" (e.g. One explanation could be that the GST tag affects the stability of the proteins).*

We agree with the reviewer that it is difficult to compare the sharpness of the centromere spots and that the word “sharp” may not be appropriate. To address this issue, we added a supplementary figure (Figure 8—figure supplement 1) where we present five representative images from EGFP-M18BP1(141-1132) and GST-EGFP-M18BP1(141-1132) cells side by side. This facilitates the comparison of the cellular distribution of EGFP-M18BP1(141-1132) and GST-EGFP-M18BP1(141-1132). These images clarify that GST-EGFP-M18BP1(141-1132) localizes more specifically to centromeres than EGFP-M18BP1(141-1132). We also define this localization as “more specific” instead of “sharper” in the main text.

Reviewer #2:

*[…] There are however some specific issues regarding the interpretation of some of the data that need to be resolved:*

*1) The title and text speak of oligomerization of M18BP1. I don´t see any evidence for this. The authors show that the Mis18α/β subcomplex can bind two molecules of an N-terminal M18BP1 fragment. This does not in itself indicate oligomerization which implies a multimerization (i.e. more than two). Using the term "Mis18α/β-induced dimerization of M18BP1" seems to me more appropriate.*

We agree with the reviewer that using the word “dimerization” is more accurate and we have changed the title accordingly.

2) Subsection “M18BP11–140 contains two sequential binding regions for Mis18α:Mis18β”:

*By identifying two consecutive fragments on M18BP1 that both retain the ability to interact with the Mis18α/β proteins does not necessarily define "sequential binding regions" or a "bipartite" (first paragraph) binding site exists as interpreted by the authors. Because the two fragments are adjacent to one another these results are consistent with a single continuous binding interface that is artificially broken into two, each retaining some binding activity. Calling this a bipartite binding site is misleading and should not be described as such.*

We have removed reference to a bipartite binding site and we only describe the binding data without trying to frame them in a precise definition.

*3) Mis18αβ stoichiometry:*

*This manuscript presents biophysical data on the stoichiometry of the Mis18α/β interpreted to be 4xalpha with 2xbeta subunits. As indicated, this is inconsistent with a previous report. The authors explain this discrepancy by the fact that they co-express the α and β subunit while in the Nardi et al. study (Nardi et al., 2016), α and β are expressed and purified separately and then mixed in a 1:1 ratio.*

*While this is a possible explanation this does not mean the experiment by the authors is superior or more likely to be correct. Given that Mis18α and β can be expressed separately and able to form a complex indicates that the stoichiometry can be established dynamically from preexisting components and does not need to be established co-translationally.*

*Further, what is the relative expression level of Mis18α and β in the insect cells used by the authors? Can the skewed α/β ration be a consequence of relative over expression of Mis18α?*

*At least, the authors should leave open the possibility that the hexamer is one likely conformation but perhaps, depending on the context, other conformations may exist.*

In this revised version of our manuscript, we have significantly extended our analysis of the subunit stoichiometry of the Mis18 complex. On the basis of this analysis, we conclude that the Mis18⟨:Mis18® does not take the form of a tetramer under any of the new conditions we have investigated, including conditions that recapitulate the previous experiments of Nardi and colleagues cited by the reviewer.

Specifically:

A) Our original studies were carried out with protein purified from insect cells after co-expression of the ⟨ and ® subunits. We now report expression of isolated 6His-Mis18⟨ and 6His-Mis18® in *E. coli* cells. Both subunits were expressed but insoluble. However, we were able to purify Mis18⟨ and Mis18® in *E. coli* by co-expression, or through addition of 6His-MBP tags to increase their solubility. Removing the 6His-MBP tag before mixing 6His-MBP-Mis18⟨ and 6His-MBP-Mis18® caused both proteins to precipitate, but if the tags were removed after mixing, the complex remained soluble. We show that *E. coli*-expressed Mis18⟨:Mis18® complex, whether produced by co-expression or by mixing of individual subunits, has essentially the same hydrodynamic profile of the insect cell sample (by gel filtration and AUC), corresponding to a hexamer. These new analyses provide rather incontrovertible evidence that the dominant form of the Mis18⟨:Mis18® complex is the hexamer and that there are no other known conditions in which a tetramer might appear.

B) We have now characterized the mechanism of hexamer formation. In agreement with Subramanian et al., 2016, the Mis18⟨:Mis18® Yippee domain form hetero-dimers and the Mis18⟨ Yippee domain also form homo-dimers. In addition, we show by AUC experiments that the Mis18⟨:Mis18® C-terminal helices form trimers containing one ® and two ⟨ subunits. These results strongly argue against the mechanism proposed by Nardi et al. that Mis18⟨ and Mis18® form tetramer via the interaction of their C-terminal helices. These experiments, shown in Figure 6, now allow us to propose a detailed model of the hexamer.

C) In addition, we now show the Mis18⟨:Mis18® Yippee domain heterodimer is where M18BP1 binds, explaining the 4:2:2 stoichiometry.

D) We kindly object to the statement that determining mass through sedimentation equilibrium and velocity “does not mean…superior or more likely to be correct”. To our knowledge, sedimentation equilibrium is superior to any other technique for mass determination, as it is completely independent of macromolecular shape and does not rely on any form of calibration, a step that is of course highly likely to introduce artifacts, in particular if performed with a sample as heterogeneous as the one used by Nardi and colleagues. On this particular point, the reviewer is referred to our answer to reviewer 3, where we provide additional evidence that the technical quality of the experiments by Nardi et al. is not comparable to that presented here.

*4) Figure 2, Figure 4, Figure 6, Figure 2—figure supplement 1:*

*Why normalize to CREST signals? This is likely to create normalization artifacts as CREST signals appear to be rather poor with high background. Further, CREST stains a region that is broader than the CENP-A domain due to CENP-B staining further complicating normalization. Finally, and importantly, most CREST antibodies recognize, in part, CENP-A which means that the changes in CENP-A signal also potentially affect the normalization signal. CENP-A-SNAP values should simply be background corrected but not normalized.*

We agree with the reviewer that the CREST signal in the blue channel is rather poor and shows high background. This is mainly due to the anti-human DyLight 405-conjugated secondary antibody we used. We also take the point that the normalization strategy we had used was not ideal. We now use the CREST signal only to determine the position of centromeres, and quantified all the images again without normalization against CREST. All images in this revised version have been re-quantified. Importantly, the new quantification does not change any of our original conclusions.

*5) Figure 2, Figure 4, and Figure 6:*

*Box plots are shown of centromere signal values. The spread in this data represents the biological variance in centromere intensities not the experimental variance. How often are these experiments repeated? In order to make any claim regarding differences and their significance (as in Figure 6) it is meaningless to compare the variance in centromere intensities. When enough centromeres are sampled there will always be a significance difference. What matters is the variance in the experiment itself. i.e. are the means of replicate experiments significantly different? Without this, these quantifications are not informative.*

We thank the reviewer for this important suggestion. We have carried out three independent replicas of the CENP-A loading experiments. In our original quantification, we treated images from independent replicas as part of the same dataset. To address the reviewer’s concern, we have re-quantified each replica separately, and the new graphs present the mean values from three independent experiments. For each replica, a mean value of fluorescence intensity was obtained from >100 centromere spots from 10-12 early G1 cells. Error bars indicate SEM. Student’s t tests were performed using the new quantifications.

*6) Final section:*

*The argument of oligomerization and its benefit to M18BP1 recruitment is weakly supported by data. Essentially, the only evidence in support of this is the slight increase in M18BP1 localization when it is forced to dimerize by fusion to GST in the absence of its ability to dimerize via Mis18 proteins (by deletion of the N-terminal 140aa).*

*This experiment is however, poorly controlled. First, as I outlined above the level of significance of the difference in the signal intensities cannot be determined by comparing the variance in centromeres across cells in a signal experiment. Instead, the reproducibility of the difference between experiments should be tested. Secondly, it is not clear from this experiment whether the perceived increase in M18BP1 centromere localization by GST dimerization compensates for the lack of dimerization via the Mis18 proteins. In other words, does GST play a role here that would otherwise be provided by the Mis18 proteins? One way to test this is to fuse GST to full length M18BP1 and test its effect on localization. In this case, if the authors´ hypothesis is correct, then M18BP1 dimerization via M18alpha and beta is functional and further dimerization via GST would have no further contribution.*

We thank the reviewer for making this important point. We agree that our experiment should include a control using GST-EGFP-M18BP1(1-1132), and we have made a HeLa CENP-A-SNAP cell line +GST-EGFP-M18BP1(1-1132)-P2AT2A-mCherry-Mis18α and performed CENP-A loading experiment as we did for other cell lines. GST-EGFP-M18BP1(1-1132) showed centromere localization as EGFP-M18BP1(1-1132) and was similarly able to rescue CENP-A loading and Mis18α localization under the depletion of endogenous M18BP1. Representative images and the quantification of the fluorescence intensity at centromeres are presented in Figure 8.

As described above in the response to the reviewer’s major point 5, we have re-quantified the localization data and applied Student’s t test again using the mean values from three independent experiments. This indicated a significant difference (P=0.0016,**) of the centromere EGFP-fluorescence intensity between cell lines expressing GST-EGFP-M18BP1(141-1132) and EGFP-M18BP1(141-1132), which is presented in Figure 8. Additionally, we included a supplementary figure (Figure —figure supplement 1) to present five representative images from both EGFP-M18BP1(141-1132) and GST-EGFP-M18BP1(141-1132) cells side by side, which makes the more robust centromere localization of GST-EGFP-M18BP1(141-1132) in comparison to EGFP-M18BP1(141-1132) easier to grasp.

We also generated four additional HeLa cell lines co-expressing following tagged proteins:

1. EGFP-M18BP1(1-140) and mCherry-M18BP1(1-140)

2. EGFP-M18BP1(1-140/T40D/S110E) and mCherry-M18BP1(1-140/T40D/S110E)

3. GST-EGFP-M18BP1(1-140) and GST-mCherry-M18BP1(1-140)

4. GST-EGFP-M18BP1(1-140/T40D/S110E) and GST-mCherry-M18BP1(1-140/T40D/S110E)

These cells were treated with CDK1 inhibitor RO-3306 and lysed for co-IP experiments using anti-GFP antibody coupled beads. With these co-IP experiments, we showed that EGFP-M18BP1(1-140) only interacts with mCherry-M18BP1(1-140) when they can bind Mis18beta, supporting our model of M18BP1 dimerization via Mis18αlhpa:β complex. We also show that GST-EGFP-M18BP1(1-140/T40D/S110E) interacts with GST-mcherry-M18BP1(1-140/T40D/S110E) even it could not bind Mis18beta, confirming the dimerization of GST-tag in HeLa cells. These new results are shown in Figure 7.

*Reviewer #3:*

*Centromere identity is epigenetically specified by nucleosomes in which H3 has been substituted for the centromere-specific histone H3 variant CENP-A. Faithful, site-specific transmission of CENP-A chromatin from one generation to the next is essential for chromosome segregation and long-term genome stability. This process is carried out by the CENP-A specific chaperone HJURP which is targeted to the proper locus by the Mis18 complex (Mis18α, Mis18β, and M18BP1) immediately following mitosis during early G1. Cell cycle regulated assembly and centromere recruitment of the Mis18 complex is essential for centromere inheritance, and a major focus of recent work in the centromere field has been to understand the basis of this regulation.*

*Previous studies have shown that Cdk phosphorylation prevents premature recruitment of the Mis18 complex to centromeres during G2 and mitosis (Silva, et al. 2012). This was extended by the demonstration that Cdk phosphorylation of M18BP1 prevents recruitment of Mis18α/β to the N-terminal 383 amino acids of M18BP1 (McKinley and Cheeseman, 2014). In this manuscript, Pan, et al. show that of the residues mutated by McKinley and Cheeseman, two of them in particular, T40 and S110, inhibit the direct interaction of M18BP1 with Mis18α/β when phosphorylated.*

We thank the reviewer for these comments. While we appreciate the importance of previous work in this area, we would also like to point out that the paper of McKinley and Cheeseman (2014) studied a construct comprising residues 1-490 of M18BP1, and that these authors mutated 18 possible phosphorylation sites to Ala (M18BP1-CDK-A), without characterizing their function in detail, as it is natural given the entity of the effort. Our study focuses on a much smaller region, defining its mechanistic significance and the role of phosphorylation, making what we think is a significant and rigorous contribution to the field.

*M18BP1 must associate with Mis18α/β to target the complex to G1 centromeres thus it is important to understand how the complex is assembled. Previous work using size exclusion chromatography and glycerol gradient sedimentation led to the proposal that the Mis18 complex comprises a Mis18 heterotetramer (2α:2β) formed by their C-terminal coiled-coil domains bound to an unknown number of M18BP1 molecules (Nardi, et al. 2016).*

This may read fastidious, but we think that it was Stellfox et al.2016 rather than Nardi et al. 2016 who showed Mis18⟨:Mis18® complex binds directly to (a significantly larger segment of) M18BP1.

*In this work, Pan, et al. use sedimentation velocity and equilibrium ultracentrifugation to determine that the Mis18 complex comprises a Mis18 heterohexamer (4α:2β), which in turn can bind 2 copies of the M18BP1 N-terminus. The authors argue that a Mis18 hexamer is the true stoichiometry because the methods used in the previous study are more error-prone. In addition, they suggest that the previous study may have obtained a 2:2 stoichiometry because they combined a 1:1 molar ratio of separately purified Mis18α and Mis18β rather than coexpressing them in eukaryotic cells.*

*The major scientific contributions in this manuscript are 1) the biochemical characterization of the Mis18 heterohexamer which clarifies previous studies and 2) the specific M18BP1 phosphosites that regulate Mis18α/β binding. Although these contributions are worthwhile and the biochemical studies are well done it is not clear whether this work warrants publication in eLife given the relatively modest advance in our understanding of Mis18 complex function in centromere formation.*

The type of thorough and detailed work presented in this manuscript tends at time to be underappreciated by editors or reviewers on the basis that it lacks conceptual novelty. We would like to argue that this type of work is instead fair rewarding because it will provide durable, trustable information to a field that has still much to discover. In this revised version of the manuscript, we have further extended our characterization of the Mis18 complex, defining in quite some detail the basis for its oligomerization and of its interaction with M18BP1. We demonstrate that M18BP1 forms dimers or oligomers via the Mis18 complex, and that this interaction is essential for CENP-A loading, because it cannot be recapitulated simply by forcing dimerization of M18BP1 via GST.

*1) The reason for the discrepancy between the stoichiometry measured by Pan, et al. and Nardi, et al. remains unclear. By size exclusion, Nardi et al. also appear to observe a mono-dispersed species with a migration consistent with their calculated molecular weight.*

We disagree with the reviewer that Nardi et al. observed mono-dispersed species. As shown in Figure 1 from Nardi et al., the one on which their hydrodynamic analysis is based, the authors only showed Western blots (WBs) of fractions from size-exclusion chromatography separations, but never showed chromatograms, nor did they show Coomassie-stained SDS-PAGE separations. It is apparent from these WBs that the indicated species often sediment or elute in more than one peak. It is our view that this evidence is incompatible with the reviewer’s conclusion that the elution profiles shown by Nardi et al. represent mono-dispersed species. We also argue that putting confidence on these profiles as a source of values for the sedimentation coefficient and the Stokes’ radius required to solve the Siegel-Monty equation (the approach used by Nardi et al.) may be a source of errors. In frankness, we hope that the reviewer will recognize that our approach cannot and should not be equaled to that used by Nardi and coworkers. We are confident that the new evidence we provide in this revised version of the manuscript will convince the reviewer.

*Pan, et al. argue based on their analyses that the Mis18 complex adopts an elongated conformation, leading to its earlier elution in size exclusion chromatography. Therefore, if anything I would expect a molecular weight calculation based on SEC elution of this complex to overestimate rather than underestimate its stoichiometry. In the absence of data supporting the idea that the stoichiometry of the complex is 'titratable'.*

The reviewer should note that the equilibrium sedimentation experiments were carried out at five different concentrations (6, 4, 2, 1, and 0.5 µM) without any evidence of changes in the stoichiometry of the complex. As noted in our response to reviewer 2, equilibrium sedimentation analysis is superior because it is entirely shape independent and provides a direct measurement of mass. As explained in our response to reviewer 2, we have now extended our analysis to a version of the complex expressed in and purified from *E. coli* and show that it has the same hydrodynamic properties, at two different concentrations, of the complex purified from insect cells.

*I also don't find the notion that Nardi et al. observed a tetramer simply because they mixed Mis18α and Mis18β in a 1:1 ratio compelling. If the ratio were off, presumably Nardi et al. would have seen a population at equilibrium containing some hexamer and some monomer/dimer/trimer. The difference in observed species may likely be due to the different expression systems used (bacterial vs. eukaryotic) and resulting differences in protein modifications/folding/etc., but there may also be some interesting biology in the differences. On the other hand, elaborating on the technical differences between the methods to clarify why the Siegel-Monty method is more error-prone would be helpful.*

We agree with the reviewer that our argument to explain the discrepancy made assumptions that cannot be proven. We have therefore removed the sentences about 1:1 mixing from our Discussion. We already underlined the implicit difficulty of using Siegel-Monty with samples like the ones used by Nardi and co-workers. Co-expression is the gold standard to solve the solubility and assembly problems of macromolecular complexes. When writing our manuscript we have tried to be respectful of the biochemical work of Nardi et al., even if we felt that it does not meet the standards required to reach definitive conclusions. Importantly, we have not been able to purify 6His-Mis18⟨ or 6His-Mis18® in isolation from bacteria due to the very poor behavior of these proteins. Addition of an MBP tag partly solved the problem, as long as the tag was not cleaved off. As already referred to above, we have only been able to purify the Mis18 complex after co-expression of Mis18⟨ and Mis18® in *E. coli*. While not as stable as the complex produced in insect cells, this bacterially expressed complex behaves in gel filtration and AUC assays precisely like the sample produced in insect cells (Figure 6).

*I don't think it's specifically up to the authors in this paper to resolve the reason for this discrepancy, but given that it's not clear whether we're looking at precisely the same complex as in Nardi et al. I think it's important for the authors to provide some additional characterization of the hexamer. Is the coiled-coil domain involved in hexamerization? Do the coiled-coil mutations described by Nardi et al. affect hexamer formation? Is the hexamer comprised of a Mis18α/β tetrameric core mediated by coiled coils with additional Mis18α bound by a different mechanism (e.g., the Yippee domain, which is also used for tetramerization in SpMis18)? The Musacchio lab has made extensive use of cross-linking/mass spectrometry in the past to understand the organization of multi-protein complexes – perhaps this could shed light on differences between the Nardi and Pan Mis18 complexes.*

As already indicated above, all our new experiments, which include experiments carried out under the conditions of Nardi et al. with protein of bacterial origin, failed to identify the existence of a Mis18 tetramer. We have now included additional evidence supporting our hexamer model. For a summary, please refer to the section of the major comment #3 of the reviewer #2.

*2) Although it's clear that Mis18 complex assembly is required for its recruitment, it is not clear whether the cell cycle timing of its recruitment is strictly determined by regulating the timing of complex assembly or whether regulation of some additional interaction plays a role. Silva, et al. found that mutating 24 putative Cdk sites (including S110 but not T40) in M18BP1 uncoupled its centromere localization from cell cycle regulation, allowing its recruitment in mitosis and G2. Can this be recapitulated by the lone S110A (or T40V/S110A) mutant characterized by Pan, et al.?*

This is a question of great interest to members of the Janssen laboratory, who have already published important work on this topic and presented at recent meetings unpublished data on the role of phosphorylation in the recruitment of CENP-A loading machinery. Partly for this reason, we have decided not to analyze this question in detail. In addition, given the very extensive level of phosphorylation of M18BP1 and other proteins in the CENP-A loading pathway, we surmise that it is unlikely that preventing the phosphorylation of the two residues we have discussed will be sufficient to bypass cell cycle inhibition of CENP-A loading. Of note, the work of Janssen and co-workers focuses precisely on this point, and reached the conclusion that lack of phosphorylation at different residues (on HJURP and M18BP1) may be sufficient to prevent CENP-A deposition in G2. More pervasive phosphorylation in M-phase may extend the regulation of additional sites, such as the ones we have identified.